# Synergizing Understanding and Generation with Interleaved Analyzing-Drafting Thinking

**Shengqiong Wu**[1,2*], **Bobo Li**[1], **Xinkai Wang**[1†], **Xiangtai Li**[3], **Lei Cui**[4], **Furu Wei**[4],
**Shuicheng Yan**[1], **Hao Fei**[1‡], **Tat-seng Chua**[1]
[1]National University of Singapore, [2]University of Oxford,
[3]Nanyang Technological University, [4]Microsoft Research

## Abstract

Unified Vision–Language Models (UVLMs) aim to advance multimodal learning by supporting both understanding and generation within a single framework. However, existing approaches largely focus on architectural unification while overlooking the need for explicit interaction between the two capabilities during task solving. As a result, current models treat understanding and generation as parallel skills rather than synergistic processes. To achieve real *synergy*, we introduce the interleaved Analyzing–Drafting problem-solving loop (**AD-Loop**), a new think paradigm that dynamically alternates between analytic and drafting operations. By interleaving textual thoughts with visual thoughts, AD-Loop enables models to iteratively refine both comprehension and outputs, fostering genuine synergy. To train this mechanism, we design a two-stage strategy: supervised learning on interleaved thought data to initialize alternation, followed by reinforcement learning to promote adaptive and autonomous control. Extensive experiments demonstrate that AD-Loop consistently improves performance across standard benchmarks for both understanding and generation, with strong transferability to various UVLMs architectures. Visual analyses further validate the effectiveness of implicit visual thoughts. These results highlight AD-Loop as a principled and broadly applicable strategy for synergizing comprehension and creation. The project page is AD-Loop.io.

## 1 Introduction

Unified Vision–Language Models (Wang et al., 2024; Wu et al., 2025b; Deng et al., 2025; Wu et al., 2025a; Xie et al., 2025) can handle both understanding and generation tasks, attracting significant research attention as they hold the potential to move beyond task-specific solutions toward general multimodal intelligence. Recent advances in multimodal large language models (MLLMs) (Liu et al., 2024a; Bai et al., 2023; Dai et al., 2023) and diffusion-based generative transformers (DiTs) (Podell et al., 2023; Yang et al., 2024; AI, 2025) have substantially improved performance in both multimodal comprehension and content creation. Building upon these advances, early efforts (Shen et al., 2023; Wu et al., 2024c) have attempted to integrate existing understanding and generation models within a single framework, enabling both capabilities simultaneously. However, such straightforward integration merely co-locates the two paradigms without enabling genuine interaction or mutual reinforcement. In essence, understanding and generation are inherently complementary (Paivio et al., 2006; Ellamil et al., 2012). Robust understanding provides the semantic foundation for faithful generation, while successful generation results can serve as tangible evidence of visual comprehension. Therefore, an effective unified model should not only combine the two capabilities but also establish a mutually reinforcing loop.

---

[*]Email: swu@u.nus.edu
[†]Work done during a remote internship at NUS.
[‡]Corresponding author: Hao Fei; Email: haofei7419@gmail.com

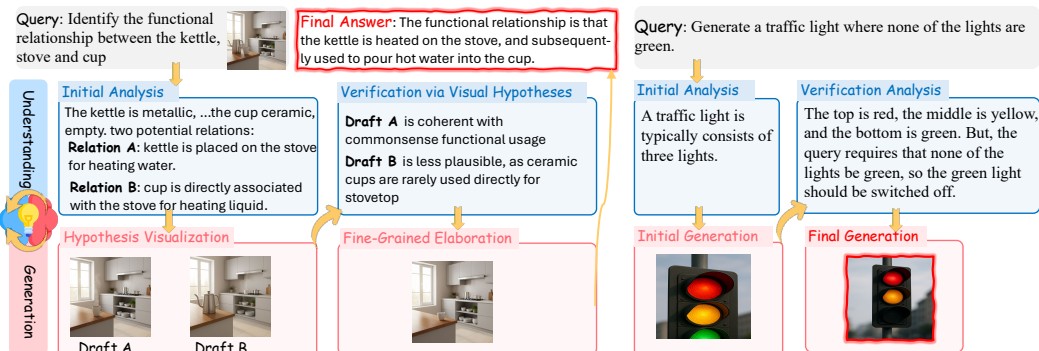

Figure 1: Illustration of the interleaved analytic–drafting problem-solving loop, where understanding and generation interact synergistically to yield accurate solutions.

To realize such synergy, early works have sought to build coherent unified architectures, for instance by casting both understanding and generation as autoregressive (AR) next-token prediction (Zhan et al., 2024; Wu et al., 2025a; Wang et al., 2024; Team, 2024a). While this design provides a unified interface from an engineering standpoint, it often suffers from issues such as *parameter competition* and *task interference*. More recent efforts explored (i) decoupled encoders with multi-head outputs to reduce representational conflicts (Chen et al., 2025c; Ma et al., 2025; Kou et al., 2024; Deng et al., 2025), and (ii) hybrid AR–diffusion models (Xie et al., 2025; Zhou et al., 2024; Xiao et al., 2025b) that take advantage of efficiency with high-fidelity rendering. Overall, these explorations of synergy have predominantly focused on improving architectural design to unify the two abilities, while overlooking a crucial point: during task solving, the understanding and generation modules often lack *close and explicit interaction*, thereby failing to realize genuine synergy between comprehension and generation. To illustrate this, when a user's instruction is ambiguous, an understanding module could first propose several plausible candidate solutions for a question, then invoke generation to produce sketches or key visualizations as a means of "verifying" these candidates, ultimately yielding the correct answer. Conversely, once the generation module produces initial results, it could query the understanding module for high-level guidance, such as attributes or reasonable spatial layouts, to progressively refine the output (see Fig. 1). This motivates a new perspective: *instead of treating understanding and generation as co-existing skills, we argue they should be interleaved in a problem-solving loop*.

To combat this, we propose a novel thinking paradigm, termed the interleaved **analyzing–drafting problem-solving loop (AD-Loop)**. The core idea is to enable the model to dynamically alternate between understanding and generation, thereby fostering an organic synergy that enhances overall problem-solving capability. Specifically, building upon existing UVLMs, we design to interleave textual thoughts, such as semantic abstraction and reasoning, with visual thoughts, including sketching and spatial layout. This dynamic switching between analytic and synthetic modes enables the model to iteratively refine its reasoning and outputs. Furthermore, we introduce a two-stage training strategy to effectively optimize the model, as shown in Fig. 2. **In the first stage**, we construct supervised datasets of interleaved textual–visual thoughts, which initialize the model with the ability to alternate between analyzing and drafting in a guided manner. **In the second stage**, we employ reinforcement learning with hybrid feedback to encourage the model to intelligently and autonomously decide when to invoke understanding versus generation, thereby fostering adaptive and self-directed synergy. Importantly, our proposed training framework and problem-solving paradigm are architecture-agnostic, making them broadly applicable to a wide range of existing UVLMs. By integrating interleaved analyzing and drafting, our approach substantially enhances their ability to achieve deep synergy between comprehension and creation.

We conduct extensive experiments to evaluate the effectiveness of the proposed AD-Loop. First, across widely used understanding and generation benchmarks, our method consistently delivers broad and significant performance improvements, including an average $+2.3\%$ improvement on understanding tasks and an overall score of $86\%$ on GenEval. Next, ablations across thinking strategies demonstrate that interleaving analytic and drafting thinking yields clear advantages. We then conduct transfer studies across different UVLM architectures, showing that our approach can be seamlessly applied to diverse models and significantly enhance both understanding and generation benchmarks compared to the original baselines. Furthermore, visualizations of implicit visual thoughts

confirm the rationality of our design. Finally, through detailed case analyses, we present compelling evidence that our model alternates between understanding and generation during problem-solving, leading to superior outcomes. Our contributions can be summarized as follows:

- We propose a fundamentally new strategy for synergizing understanding and generation by introducing the interleaved analyzing–drafting thinking mechanism, which enables tight, explicit interactions between the two capabilities.
- We develop a novel two-stage learning strategy that equips UVLMs with the ability to intelligently adopt interleaved analytic–drafting problem-solving loops during the task-solving process. Moreover, the framework is architecture-agnostic across different UVLMs.
- Extensive experiments on understanding and generation benchmarks demonstrate the effectiveness of our approach, providing intuitive visual analyses that confirm both the rationality and efficacy of the proposed synergy.

## 2 RELATED WORK

The mutual benefits between understanding and generation have long been a central theme (Paivio et al., 2006; Ellamil et al., 2012; Fei et al., 2025). Enabled by rapid progress in multimodal large language models (MLLMs) (Liu et al., 2024a; Bai et al., 2023; Li et al., 2024; Fei et al., 2024) and diffusion-based generators (Podell et al., 2023; Yang et al., 2024), state-of-the-art systems now achieve strong performance on each task in isolation. This has sparked growing interest in unified vision–language models (UVLMs) (Xie et al., 2025; Kou et al., 2024; Wu et al., 2025b), which support multimodal understanding and generation within a single framework. Early explorations (Shen et al., 2023; Wu et al., 2024c) largely composed powerful specialist models by directly connecting an understanding model with a generation model. However, such plug-and-play integration merely co-locates the two capabilities and does not realize mutual assistance. More recent efforts pursue more coherent formulations that jointly model both tasks, e.g., casting both as autoregressive next-token prediction (Wu et al., 2025a; Team, 2024a; Chen et al., 2025c;b; Geng et al., 2025b) or adopting unified AR-diffusion architectures (Xie et al., 2025; Zhou et al., 2024; Deng et al., 2025). Yet even these tighter designs still treat understanding and generation as parallel, independently callable skills; at inference time, the two modules rarely engage in close, explicit interaction, and genuine synergy remains elusive. In contrast, we introduce a novel thinking paradigm that enables true synergic learning between understanding and generation through an interleaved analyzing–drafting loop.

A complementary line of work (Shao et al., 2024; Xu et al., 2024; Shen et al., 2025; Wang et al., 2025a) focuses on endowing MLLMs with strong reasoning capabilities. Early approaches follow the Chain-of-Thought paradigm (Wei et al., 2022) by decomposing problems into substeps and supervising stepwise rationales. Subsequent methods scale test-time computation (Snell et al., 2024) via self-consistency or majority voting and incorporate search-based inference (Yao et al., 2024), while training-time credit assignment is improved through reinforcement learning (RL) or preference optimization tailored to reasoning. More recently, RL has been explored to elicit emergent reasoning abilities that improve multimodal understanding (Huang et al., 2025; Chen et al., 2025a; Yuan et al., 2025) and generation (Xiao et al., 2025b; Jiang et al., 2025a). In light of this, some explorations (Jiang et al., 2025b; Yan et al., 2025) attempt to leverage RL to co-optimize understanding and generation, thereby strengthening their respective capabilities. Beyond textual-only reasoning thoughts, several studies (Cheng et al., 2025a; Shao et al., 2024) interleave textual reasoning with visual evidence, either via tool-augmented pipelines (Zheng et al., 2025; Zhang et al., 2025a; Man et al., 2025; Su et al., 2025) or by directly generating visual traces (Chern et al., 2024; Li et al., 2025; Yang et al., 2025b). Building on these advances, our work models the synergy between understanding and generation as an interleaved analyzing–drafting problem-solving loop that jointly produces textual and visual thoughts, thereby enhancing synergic learning within UVLMs.

## 3 METHODOLOGY

Building upon a UVLM, we model the synergetic *understanding* and *generation* thinking process as an **interleaved analyzing–drafting problem-solving loop**. Given an input, the model alternates between analyzing (producing *textual thoughts*) and drafting (producing *visual thoughts*) before delivering the final output. To achieve this, we design a two-stage training pipeline: **Stage 1** performs supervised training to *imitate interleaved thinking*, and **Stage 2** leverages reinforcement learning to enable the model to *adaptively decide when to invoke* analysis or drafting.

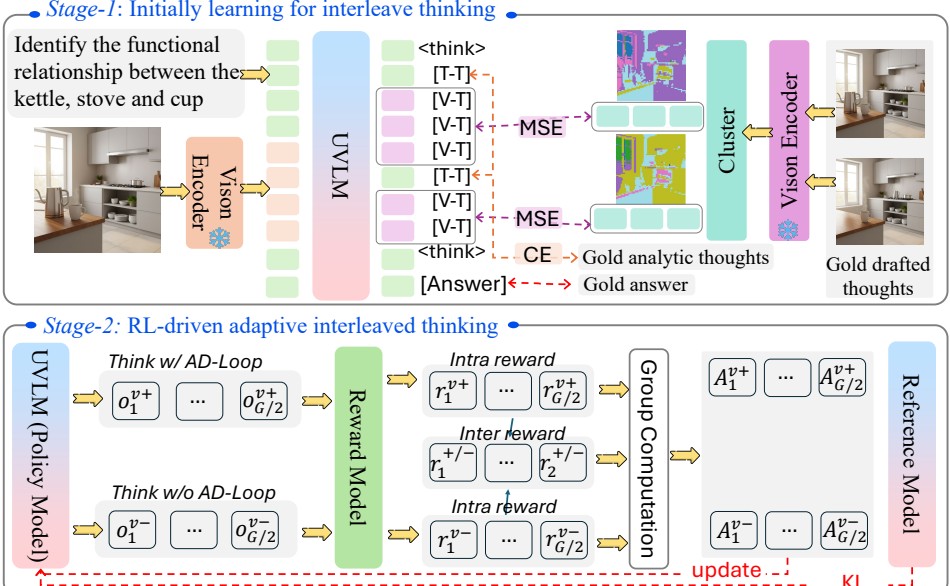

Figure 2: Pipeline of our training framework. **Stage-1**: train the UVLM to emit interleaved thinking traces. **Stage-2**: apply GRPO for hybrid reasoning. The policy samples multiple traces with/without the interleaved AD-Loop for each input. A reward model scores outcomes, and then group-normalized advantages are applied to update the policy, teaching the model when AD-Loop helps.

## 3.1 MODEL ARCHITECTURE

Let the input be $x = (q, \mathcal{I})$, where $q$ is text and $\mathcal{I} = \{I_m\}_{m=1}^M (M \geq 1)$ is an optional set of images. The input image set $\mathcal{I} = \{I_m\}$ is encoded by a vision encoder into visual embeddings, which are fused with the textual stream and then processed by an LLM backbone for reasoning. The model outputs a thinking trace wrapped in `<think>` and `</think>` tags and a final outcome:

$$\texttt{<think> [T-T] [V-T] [T-T] [V-T] ...</think> [Answer]}, \qquad (1)$$

where `[T-T]` denotes *text thought*, `[V-T]` denotes *visual thought*, which is enclosed by two special tokens marking the beginning and end of the visual thought. `[Answer]` is the final text or image. For image synthesis, prior unified models typically adopt either (i) **discrete** codebooks that predict quantized "image tokens" (Chen et al., 2025c; Xie et al., 2025), or (ii) **continuous** latent regressors that predict low-dimensional vectors consumed by a renderer (Deng et al., 2025). Emitting a *full* visual scene during the thought phase requires long token streams (often hundreds of tokens or prolonged diffusion steps), which increases latency and entangles reasoning with pixel-level details that are irrelevant to the decision. Motivated by cognitive accounts that internal imagery is schematic rather than pixel-accurate (Shepard & Metzler, 1971), we replace full rendering during thinking with a compact set of *latent visual thoughts*, $\{v_j\}_{j=1}^K$, which summarizes only the factors *useful for reasoning* under a strict budget $K \ll$ the tokens required to render an image. This design preserves the sufficiency of visual cues for the downstream decision while avoiding the cost of pixel-level synthesis in the thought process.

## 3.2 STAGE-1: SUPERVISED TRAINING OF INTERLEAVED THINKING

We begin with supervised fine-tuning on interleaved reasoning data. This approach mitigates the instability commonly observed in cold-start RL for reasoning and provides a strong initialization for next-stage reinforcement learning.

**Dataset Construction.** Our interleaved corpus comprises two halves: *i) Understanding.* We leverage the existing multimodal CoT resources (Cheng et al., 2025b; Zhang et al., 2025b; Shao et al., 2024), reorganizing their rationales and cropping the referenced regions according to the provided annotations to construct the interleaved schema. We further synthesize AD-Loop thinking traces by instantiating schematic grid tasks following Li et al. (2025). In total, the understanding portion contains `20K` interleaved examples. *ii) Generation.* We use the GoT dataset (Fang et al., 2025) and

additionally construct an Inter-T2I set from X-to-image corpora (Xiao et al., 2025a), augmenting each instance with an interleaved thinking trace. This yields `22K` interleaved generation traces.

**Training.** To teach the model an AD-Loop style of thinking, we fine-tune the UVLM directly on the above collected corpus, as shown in Fig. 2. A practical issue is that the data sources provide explicit visual thoughts (i.e., pixel images), whereas our reasoning format requires latent visual thoughts. We therefore convert each explicit visual-thought image $I$ into a compact set of tokens via a *frozen* generator-side encoder and a deterministic clustering step. Technically, we reuse the encoder for image generation from the unified model. Given $I$, the encoder yields a grid of latent tokens $\{z_i\}_{i=1}^N$. Rather than supervising the model to emit the full grid, we reduce $\{z_i\}_{i=1}^N$ to a small, semantically coherent set $\{v_j\}_{j=1}^K$ with $K \ll N$. Following Jin et al. (2024), we adopt a density peaks clustering mechanism to construct the token clusters. For each cluster $\mathcal{C}_j$, we produce a representative latent visual thought token $v_j$ as the union of members $v_j = \frac{1}{|\mathcal{C}_j|}\sum_{i\in\mathcal{C}_j} z_i$. The clustering mechanism yields stable targets that merge tokens based on semantic proximity in the latent space, rather than through naive spatial pooling. We order $\hat{\mathcal{V}} = \{v_j\}_{j=1}^K$ by the center coordinates of their clusters (top–left to bottom–right) to obtain a deterministic sequence. Let $\mathcal{T}^\star$ and $\mathcal{V}^\star$ denote the gold text-thought and visual thoughts sequences in the `<think>` block, and let $o^\star$ denote the final output. The final training objective is to optimize:

$$\mathcal{L}_{\text{S1}} = \underbrace{\mathcal{L}_{\text{CE}}\left(\hat{\mathcal{T}}, \mathcal{T}^\star\right)}_{\text{text thoughts}} + \alpha\, \underbrace{\mathcal{L}_{\text{vis}}\left(\hat{\mathcal{V}}, \mathcal{V}^\star\right)}_{\text{latent visual thoughts}} + \underbrace{\mathcal{L}_{\text{out}}(\hat{o}, o^\star)}_{\text{task output}}, \tag{2}$$

where $\mathcal{L}_{\text{vis}}$ is mean-squared error, and $\mathcal{L}_{\text{out}}$ is the original task loss. $\alpha$ is the coefficient weight.

### 3.3 STAGE-2: RL-DRIVEN ADAPTIVE INTERLEAVED THINKING

After Stage–1 fine-tuning, the model has acquired an initial interleaved thinking capability. However, some queries might be solved confidently with an isolated thinking mechanism, i.e., invoking only the understanding or generation capability. Accordingly, Stage-2 further strengthens the model's thinking competence and makes the policy *adaptive* for each input. The model should be able to decide whether to think with AD-Loop ($\mathcal{V}+$) or without it ($\mathcal{V}-$).

Drawing on group-relative preference optimization (Guo et al., 2025; Jiang et al., 2025c), we introduce an *adaptive interleaved thinking* regimen, as illustrated in Fig. 2. For every query, the policy explores both thinking modes and receives inter- and intra-group normalized feedback. The resulting group-relative advantages drive on-policy updates toward a parsimonious strategy. We next detail the sampling scheme in *Adaptive Interleaved Thinking*, the reward design in *Reward Assignment*, and the update rule in *Policy Optimization*.

**Adaptive Interleaved Thinking.** For each query $q$, we sample two *groups* of trajectories from the old policy $\pi_{\text{old}}$: one with the AD-Loop enabled ($\{o_i^+\}_{i=1}^{G/2}$), and one with the AD-Loop disabled ($\{o_i^-\}_{i=1}^{G/2}$). Let $G$ be the total number of samples per query:

$$\{o_i^+\}_{i=1}^{G/2} \sim \pi_{\text{old}}, \;\; \{o_i^-\}_{i=1}^{G/2} \sim \pi_{\text{old}}, \;\; O = \{o_i^+\}_{i=1}^{G/2} \cup \{o_i^-\}_{i=1}^{G/2} \tag{3}$$

**Reward Assignment.** Following Guo et al. (2025), each trajectory is assigned a scalar reward composed of a *format* component and a *content* component. For generation tasks, the content term aggregates alignment and quality scores (e.g., preference feedback (Wu et al., 2023), unified scoring (Wang et al., 2025b), or text–image alignment (Geng et al., 2025a)), while for understanding tasks, we apply rule-based checks to enforce correctness:

$$r_{base}(o) = r_{\text{format}} + r_{\text{content}}. \tag{4}$$

To explicitly encourage *useful* AD-Loop, we add a margin-based bonus to $\mathcal{V}+$ only when it outperforms the strongest $\mathcal{V}-$ candidate and the AD-Loop contributes meaningfully:

$$r(o_i^+) = r_{\text{base}}(o_i^+) + \lambda\, \mathbf{1}(\text{AD-Loop}|a)\max\left(0, r_{\text{base}}(o_i^+) - \max_j r_{\text{base}} - \delta\right), \;\; r(o_i^-) = r_{\text{base}}(o_i^-), \tag{5}$$

where $\mathbf{1}(\text{AD-Loop}|a)$ is an indicator that the answer is correct and AD-Loop thinking is employed. The margin parameter $\delta$ requires the $\mathcal{V}+$ candidate to exceed the strongest $\mathcal{V}-$ baseline by at least

Table 1: Comparison with baselines on multimodal understanding benchmarks. "Und." and "Gen." denote *understanding* and *generation*, respectively.

| Model | #Params | POPE↑ | MME-P↑ | MMB↑ | SEED↑ | GQA↑ | MMMU↑ | MM-Vet↑ |
|---|---|---|---|---|---|---|---|---|
| ● *Und. Only* | | | | | | | | |
| LLaVA-v1.5 (Liu et al., 2024a) | 7B | 85.9 | 1510.7 | 64.3 | 58.6 | 62.0 | 35.4 | 31.1 |
| Qwen-VL-Chat (Bai et al., 2023) | 7B | - | 1487.5 | 60.6 | 58.2 | 57.5 | – | – |
| IDEFICS (Laurençon et al., 2023) | 8B | - | - | 48.2 | - | 38.4 | - | - |
| InstructBLIP (Dai et al., 2023) | 13B | 78.9 | 1212.8 | - | - | 49.5 | – | 25.6 |
| ● *Und. and Gen.* | | | | | | | | |
| Emu3 (Wang et al., 2024) | 8B | 85.2 | 1244.0 | 58.5 | 68.2 | 60.3 | 31.6 | 37.2 |
| Show-o (Xie et al., 2025) | 1.3B | 80.0 | 1097.2 | – | – | 58.0 | 26.7 | – |
| Liquid (Wu et al., 2024a) | 8B | – | 1448.0 | – | – | 61.1 | – | – |
| MMaDA (Yang et al., 2025a) | 8B | 86.1 | 1410.7 | 68.5 | 64.2 | 61.3 | 30.2 | – |
| Janus-Pro (Chen et al., 2025c) | 7B | 87.4 | 1567.1 | 79.2 | 72.1 | 62.0 | 41.0 | 50.0 |
| BAGEL (Deng et al., 2025) | 7B | – | 1687.0 | 85.0 | – | – | 55.3 | 67.2 |
| **AD-Loop (Ours)** | 7B | **90.1** | **1696.0** | **87.6** | **74.4** | **63.8** | **57.3** | **69.7** |

Table 2: Comprehensive generation comparison on GenEval (Ghosh et al., 2023) benchmark. "Und." and "Gen." denote *understanding* and *generation*, respectively.

| Model | Single Obj.↑ | Two Obj.↑ | Counting↑ | Colors↑ | Position↑ | Attri.↑ | Overall↑ |
|---|---|---|---|---|---|---|---|
| ● *Gen. Only* | | | | | | | |
| Emu3-Gen (Wang et al., 2024) | 0.98 | 0.71 | 0.34 | 0.81 | 0.17 | 0.21 | 0.54 |
| SDXL (Podell et al., 2023) | 0.98 | 0.74 | 0.39 | 0.85 | 0.15 | 0.23 | 0.55 |
| FLUX.1-dev (Yang et al., 2024) | 0.99 | 0.81 | 0.79 | 0.74 | 0.20 | 0.47 | 0.67 |
| SD3-Medium (AI, 2025) | 0.99 | 0.94 | 0.72 | 0.89 | 0.33 | 0.60 | 0.74 |
| ● *Und. and Gen.* | | | | | | | |
| Show-o (Xie et al., 2025) | 0.95 | 0.52 | 0.49 | 0.82 | 0.11 | 0.28 | 0.53 |
| TokenFlow-XL (Xie et al., 2025) | 0.95 | 0.60 | 0.41 | 0.81 | 0.16 | 0.24 | 0.55 |
| Janus-Pro (Chen et al., 2025c) | 0.99 | 0.89 | 0.59 | 0.90 | 0.79 | 0.66 | 0.80 |
| BAGEL (Deng et al., 2025) | 0.99 | 0.94 | 0.81 | 0.88 | 0.64 | 0.63 | 0.82 |
| MindOmni (Xiao et al., 2025b) | **0.99** | 0.94 | 0.71 | 0.90 | 0.71 | 0.71 | 0.83 |
| **AD-Loop (Ours)** | 0.98 | **0.94** | **0.83** | **0.90** | **0.80** | **0.74** | **0.86** |

$\delta$, filtering out spurious wins and favoring the simpler text-only mode unless a meaningful gain is achieved. $\lambda$ scales this bonus, balancing its impact against the base reward. Furthermore, we introduce inter-group reward to indicate which thinking mode performs best for each query:

$$r_{\text{inter}}(o_i^m) = \begin{cases} 1, & \text{if } m = \arg\max_{m' \in \{+, -\}} \{r(o^+), \ r(o^-)\} \\ 0, & \text{otherwise} \end{cases} , \quad r_{\text{intra}}(o_i^m) = r(o_i^m), \ m \in \{+, -\} \quad (6)$$

**Optimization.** Following group-relative preference optimization, we combine an intra-group advantage with an optional inter-group term:

$$A_i = \underbrace{\left[ \frac{r_{\text{intra}}(o_i) - \text{mean}(r_{\text{intra}}(o_j))}{\text{std}(r_{\text{intra}}(o_j))} \right]}_{\text{GRPO for intra-group advantage } A_{\text{intra}}} + \gamma \underbrace{\left[ \frac{r_{\text{inter}}(o_i) - \text{mean}(r_{\text{inter}}(o_j))}{\text{std}(r_{\text{inter}}(o_j))} \right]}_{\text{GRPO for inter-group advantage } A_{\text{inter}}}, \quad (7)$$

This group-relative advantage $A_i$ of $i$-th response encourages the model to prioritize the response with higher relative quality. $\gamma$ is the weighted parameter. Following Guo et al. (2025), we apply KL-divergence regularization with a hyperparameter $\beta$ and a clipping trick to optimize the model.

## 4 EXPERIMENTS

### 4.1 SETTINGS

**Dataset.** We construct the training dataset to emerge the synergy between understanding and generation, as described in Sec. §3.2, encompassing interleaved reasoning for both understanding and generation. The detailed statistics are presented in Appendix §F.

**Implementations.** Our backbone is BAGEL-7B (Deng et al., 2025) in which SigLIP2-so400m/14 (Tschannen et al., 2025) is adopted as the image encoder for understanding, and the

Table 3: Comparison with different thinking strategies. *Isolated thinking*: understanding or generation performed in isolation; $T$: analyzing-only thinking; $T + I$ *(explicit)*: supervised textual–visual (explicit) interleaving; $T + \widetilde{I}$ *(implicit)*: learned textual–visual (implicit) interleaving; $T / T + \widetilde{I}$: adaptive interleaved thinking that automatically selects with or without interleaved thinking.

| Think Strategy | Understanding | | | Generation (WISE) | | |
|---|---|---|---|---|---|---|
| | MathVista↑ | LogicVista↑ | SAT↑ | Cultural↑ | Space↑ | Biology↑ |
| Isolated thinking | 61.5 | 40.2 | 0.63 | 0.44 | 0.68 | 0.44 |
| $T$ | 68.3 | 44.1 | 0.74 | 0.67 | 0.69 | 0.56 |
| $T + I$ *(explicit)* | 72.9 | 46.6 | 0.81 | 0.73 | 0.74 | 0.64 |
| $T + \widetilde{I}$ *(implicit)* | 73.6 | 47.2 | 0.84 | 0.75 | 0.77 | 0.65 |
| $T / T + \widetilde{I}$ | **75.8** | **49.5** | **0.89** | **0.79** | **0.78** | **0.68** |

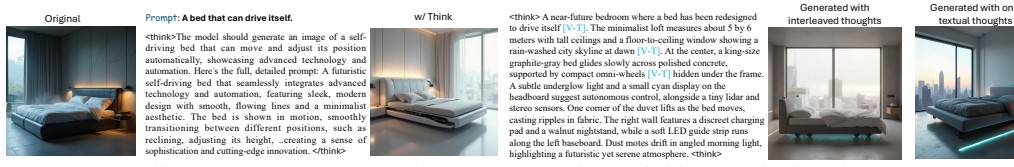

Figure 3: Qualitative comparison: original prompt (left), self-think mode, interleaved thoughts, and text-only thoughts filtered from the interleaved thoughts (right). [V-T] means latent visual thoughts.

FLUX pre-trained VAE (Yang et al., 2024) is utilized as the image latent encoder/decoder for generation. We train the model in two stages. At stage-1, we fine-tune on the curated interleaved reasoning corpus using a global batch size of 256 and a cosine learning-rate schedule with initial learning rate $1 \times e^{-5}$. The maximum number of a latent visual thought is set to $K = 16$ tokens. At stage-2, we implement RL with the VERL framework (Sheng et al., 2025). We fix the random seed to 42 to ensure reproducibility. The policy is optimized with AdamW, a constant learning rate of $2 \times e^{-6}$, a global batch size of 64, and 8 rollouts per prompt. The objective combines the clipped policy-gradient loss ($\epsilon = 0.5$) with a KL regularization to a frozen reference model weighted by 0.01.

## 4.2 MAIN RESULTS

**Understanding.** We compare the proposed method against state-of-the-art understanding-only and unified models on multimodal understanding benchmarks, including POPE (Li et al., 2023b), MME (Zhang et al., 2021), MMB (Liu et al., 2024b), SEED (Li et al., 2023a), GQA (Hudson & Manning, 2019), MMMU (Yue et al., 2024), MM-Vet (Yu et al., 2023). As highlighted in Table 1, our method consistently yields the best overall results. This improvement stems from interleaving visual-text reasoning, which effectively mitigates conflicts between understanding and generation while unlocking their synergy. Notably, even compared to MMaDA (Yang et al., 2025a), which utilizes RL-based thinking training, our approach still yields significant gains, confirming that generation can indeed enhance understanding.

**Generation.** We assess visual generation performance on the GenEval benchmark (Ghosh et al., 2023). As shown in Table 2, our method achieves the highest overall score of 86%, outperforming both generation-only and unified baselines. In particular, it delivers substantial improvements in fine-grained attributes in positional accuracy and attribute correctness. Moreover, compared with MindOmni (Xiao et al., 2025b) that employs textual thinking only, our approach achieves better performance, clearly demonstrating the effectiveness of incorporating visual thoughts.

## 4.3 ABLATION ON THINKING TYPES

We systematically evaluate reasoning strategies across both understanding and generation benchmarks, including MathVista (Lu et al., 2023), LogicVista (Xiao et al., 2024), SAT (Ray et al., 2024), and WISE (Niu et al., 2025). Results are summarized in Table 3. First, compared with the no-think setting, equipping the model with reasoning capability (*T-think*) yields substantial improvements in both understanding and generation, confirming the effectiveness of reasoning guidance. Second, augmenting textual reasoning with visual thoughts further boosts performance, highlighting that visual cues supply fine-grained details complementary to text. When comparing explicit ($T + I$-think) and implicit ($T + \widetilde{I}$-think) visual thoughts, we observe only marginal differences. This can be partly owed to the learning paradigm, and explicit supervision can introduce pixel-level noise, whereas

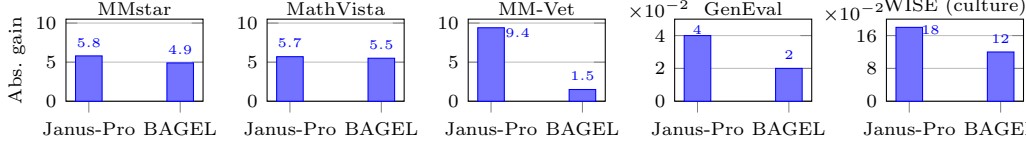

Figure 4: Absolute performance gain after applying AD-Loop, comparing Janus-Pro with discrete tokenization and BAGEL with continuous embedding for visual thoughts learning.

Table 4: Comparison of latent visual representations learn by the generation (Gen.) vs. understanding (Und.) encoder.

|  | MMstar | MathVista | LogicVista | GenEval | WISE (cultural) | WISE (Biology) |
|---|---|---|---|---|---|---|
| From Gen. Encoder | 54.9 | 75.8 | 47.5 | 0.86 | 0.79 | 0.68 |
| From Und. Encoder | 51.6 | 70.9 | 44.3 | 0.84 | 0.71 | 0.61 |

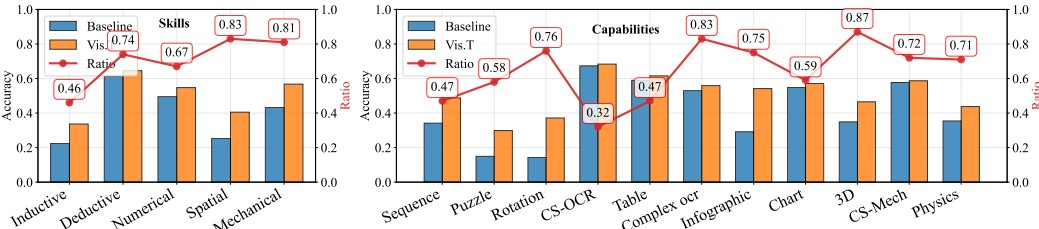

Figure 5: Examples of latent visual thoughts. Each case shows the original image (left) and the corresponding visual thoughts (right), capturing abstract visual structures.

Figure 6: Performance across skills and capabilities on the LogicVista dataset, comparing models with and without visual thoughts, alongside the proportion of visual-thought usage.

implicit reasoning captures compact salient cues. Nonetheless, combining the two in our hybrid setting achieves the best results across all tasks, suggesting a strong complementarity.

Further qualitative evidence is provided in Fig. 3. With the raw prompt alone, the model tends to generate superficial, semantically shallow outputs. Adding self-think produces more detailed descriptions, yet still overly abstract and often misaligned with user intent. By contrast, interleaved thoughts guide faithful, detail-oriented outputs (e.g., correct wheels/screens). Finally, filtering interleaved traces to text-only frequently reintroduces errors (e.g., lighting/positioning), underscoring the necessity of visual thoughts for high-fidelity controllability.

## 4.4 ANALYSES AND DISCUSSION

In this section, we are about to answer the following in-depth questions:

**RQ-1: Can this method extend into different structures of unified MLLMs?** Current unified MLLMs adopt different architectures for visual content generation. One line of work, including our backbone, generates continuous embeddings, while another, exemplified by Janus-Pro (Chen et al., 2025c), produces discrete tokens. We apply our proposed method to both paradigms, as described in Sec. §3.2. As shown in Fig. 4, Unified-R1 consistently improves both understanding and generation across the two settings. These results demonstrate that our approach is architecture-agnostic and can serve as a general mechanism for strengthening emergent task synergy in unified MLLMs.

**RQ-2: Should visual thoughts be derived from understanding or generation?** We compare visual thoughts derived from the understanding versus the generation encoder. As shown in Table 4, using the generation encoder yields consistently better results for both understanding and generation tasks. This can be attributed to two aspects. On the one hand, we observe that models converge

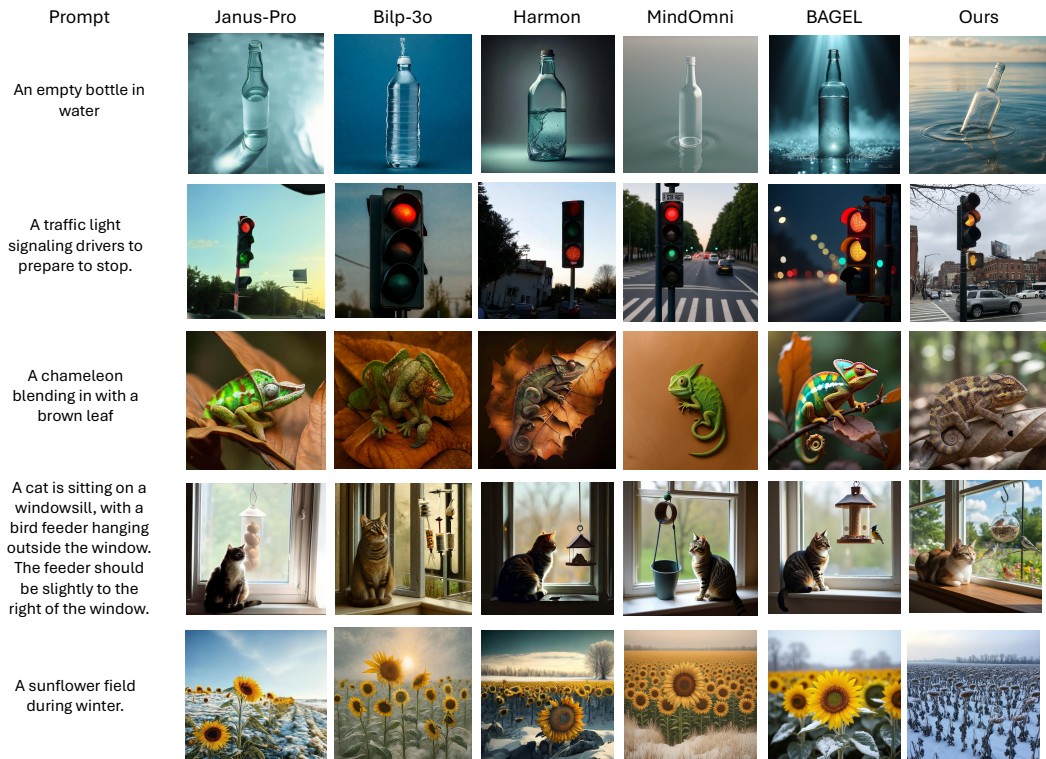

Figure 7: Comparison of unified MLLMs on T2I generation. Existing models often fail on prompts requiring deeper reasoning, whereas our approach yields more faithful outcomes.

faster under this setting, likely because generation-oriented thoughts are already extensively pretrained. Moreover, generation-based visual thoughts inherently capture both semantic and pixel-level information, making them more informative and beneficial for multimodal reasoning.

**RQ-3: What do the implicit visual thoughts look like?** We visualize the clustering results in Fig. 5. The observations align well with our expectations: latent visual thoughts encode semantically coherent information while preserving coarse pixel-level structures. This allows the model to recover the overall contours of the original image and to unify conceptually similar regions. For example, in case (4), distinct regions depicting watermelons and lemons are consistently represented by the same latent token, reflecting their shared conceptual category.

**RQ-4: When are the visual thoughts needed?** By comparing performance across different task scenarios and analyzing the proportion of visual-thought usage, we obtain the results shown in Fig. 6. Overall, integrating AD-Loop thoughts improves performance across a wide range of questions, with pronounced gains in spatial and mechanistic reasoning. Fine-grained trends show preferential activation for rotation, complex OCR, and 3D perception, while usage drops for tables, sequences, and symbolic reasoning, where text-only chains suffice. These patterns indicate our adaptive policy that selectively invokes visual thoughts when they offer the greatest benefit.

**RQ-5: Case study.** Finally, we provide qualitative analyses of our model's capabilities in text-to-image generation. As demonstrated in Fig. 7, baseline models often falter on reasoning-intensive prompts, capturing superficial cues rather than underlying logic—for example, outputting red/green for "a traffic light signaling drivers to prepare to stop" (correct: yellow), rendering a green chameleon on a brown leaf (ignoring adaptive coloration), or producing bright sunflowers for "a sunflower field during winter" (ignoring seasonality). In contrast, our method consistently aligns outputs with prompt semantics and the required reasoning. On the understanding side (Fig. 10), our model exhibits stronger spatial reasoning, correctly localizing objects and motions where baselines fail to do so. These cases highlight how interleaved reasoning facilitates faithful, reasoning-aware generation and a more robust understanding. Additional examples can be found in Appendix §H.

## 5 CONCLUSION

In this work, we introduce a unified interleaved thinking framework, i.e., AD-Loop, for synergizing understanding and generation in UVLMs. Our proposed two-stage learning paradigm first initializes an interleaved AD-Loop thinking through supervised fine-tuning, and then employs hybrid reinforcement learning to enable the model to adaptively invoke interleaved thinking when beneficial. This allows the model to learn *when* and *how* to leverage different capabilities, thereby unlocking genuine synergy across tasks. Extensive experiments on diverse multimodal understanding and generation benchmarks demonstrate the effectiveness of our approach, with particularly strong gains on reasoning-driven tasks. These results highlight the potential of interleaved AD-Loop as a general mechanism for advancing unified multimodal intelligence.

## ACKNOWLEDGEMENTS

This work was supported by a Senior AI 2050 Fellowship from the Schmidt Sciences Foundation, awarded to Michael Wooldridge.

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

APPENDIX INDEX

This supplementary material includes the following sections:

## A    THE USE OF LARGE LANGUAGE MODELS (LLMS)

In this paper, we employ the large language models (LLMs) to help the dataset construction and improve the clarity and readability of English writing. Specifically, we employ the QVQ-72B-Preview to construct the Inter-T2I dataset. Additionally, LLMs were utilized to refine sentence structure, correct grammatical errors, and enhance the overall presentation of our draft. The technical content, research ideas, experimental design, analysis, and conclusions were entirely conceived, implemented, and validated by the authors without reliance on LLMs.

## B    ETHICS STATEMENT

All datasets used in this work are publicly available and open-source. During the process of constructing our experimental data, we employed open-source text-to-image generation models. To mitigate risks of harmful content, discrimination, or bias, all generated samples were manually screened and filtered to ensure suitability for research purposes. Our approach is built on an open-source foundation model that provides strong assurance for generating non-harmful and bias-free content. Nonetheless, as with any generative system, it is impossible to guarantee that unintended or potentially harmful outputs will never occur. We therefore emphasize that users and practitioners should exercise caution when deploying such models in downstream applications, especially in sensitive domains.

We do not involve any human subjects, private or proprietary data, or personally identifiable information (PII) in this research. The work complies with community standards on data usage and research integrity. No legal, ethical, or security violations are posed by the methodologies or experiments presented in this paper. Furthermore, we highlight that our contributions are intended solely for academic and scientific exploration. We explicitly discourage misuse of the proposed methods for generating harmful, misleading, or discriminatory content. Future applications should integrate appropriate safeguards, fairness considerations, and content moderation mechanisms.

## C    REPRODUCIBILITY STATEMENT

To ensure the reproducibility of our work, we have made a concerted effort to provide all necessary details and materials. We provide comprehensive details of the proposed AD-Loop framework, including its definition, input–output formulation, and implementation (Section §3.1). The model backbone and training methodology are described in detail in Section §3 and Appendix §D. We further report all hyperparameter settings and training configurations in Section 4.1 and Appendix §G, using fixed random seeds to ensure the replicability of the experiments. All datasets used in this study are publicly available open-source resources, and the data construction process, along with the amount of data used at each training stage, is thoroughly documented in Appendix §F. Finally, we will release the full codebase and data processing scripts to the community upon acceptance.

## D    DETAILED METHODOLOGY

### D.1    LATENT VISUAL THOUGHTS CONSTRUCTION.

Given an image visual thought $I$, the image encoder yields a grid of latent tokens $\boldsymbol{Z} = \{\boldsymbol{z}_i\}_{i=1}^N$, where the image encoder can be either a VQ tokenizer (Sun et al., 2024; Chen et al., 2025c) or a VAE encoder (Yang et al., 2024; Deng et al., 2025). Inspired by Wu et al. (2024b), we then calculate the local density $\rho_i$ of the token $\boldsymbol{z}_i \in \boldsymbol{Z}$ by referring its neighbors:

$$\rho_i = \exp(-\frac{1}{K} \sum_{\boldsymbol{z}_m \in \text{KNN}(\boldsymbol{z}_i, \boldsymbol{Z})} ||\boldsymbol{z}_m, \boldsymbol{z}_i||^2), \tag{8}$$

where $\text{KNN}(\boldsymbol{z}_i, \boldsymbol{Z})$ denotes the $K$-nearest neighbors of $\boldsymbol{z}_i$ in $\boldsymbol{Z}$. We then measure the minimal distance $\delta_i$ between the feature $\boldsymbol{z}_i$ and other features with higher density:

$$\delta_i = \begin{cases} \min_{m:\rho_m > \rho_i} \varphi(\boldsymbol{z}_m, \boldsymbol{z}_i), & \text{if } \exists\, m, n : \rho_m > \rho_i \\ \max_m \varphi(\boldsymbol{z}_m, \boldsymbol{z}_i), & \text{otherwise} \end{cases} \tag{9}$$

In essence, $\delta_i$ represents the distance between the given token $\boldsymbol{z}_i$ and other high-density tokens. We summarize the score $s_i$ of the feature by combining the local density $\rho_i$ and minimal distance $\delta_i$ as $\rho_i \times \delta_i$. We identify those tokens with relatively high scores, $s_i$, as cluster centers and then allocate other tokens to their nearest cluster center based on the Euclidean distances. Finally, we utilize the average token within each cluster to represent the corresponding cluster. The latent visual thought token of the merged patch token is the union of the vision regions within the corresponding cluster.

### D.2    REWARD MODEL

In **Stage 2**, each model response is evaluated with two complementary signals: (i) a *format reward* that enforces structural validity (e.g., required fields, schema conformity), and (ii) a *content reward* that measures semantic fidelity and task-specific quality. Below, we describe the content rewards used for different task types.

**Understanding-Task Reward Model**    For tasks with deterministic targets, such as multiple-choice or numeric questions, we employ rule-based matching after normalization (e.g., case folding, whitespace removal, and unit normalization). This yields a precise, low-variance signal of correctness. For open-ended understanding questions, we rely on an external, learned reward model as the judge; specifically, we use InternLM-XComposer2.5-Reward (Zang et al., 2025), which scores responses by holistic relevance and coherence with the instruction.

**Generation-Task Reward Model**    For assessing the quality of generated images, we employ two complementary criteria: 1) **Semantic alignment score:** Measures agreement between the generated image and the ground-truth prompt via cosine similarity between CLIP image and text embeddings. 2) **Human preference alignment score:** Captures perceived aesthetic quality and prompt adherence using learned preference models, namely HPS v2 (Wu et al., 2023) and ImageReward (Xu et al., 2023). These signals are aggregated to provide fine-grained, content-aware feedback that guides the model toward faithful, high-quality generations while maintaining the required output format.

### D.3    THE OPTIMIZATION OBJECTIVES IN STAGE-2

After obtaining the advantages, we apply the standard objectives as described in Guo et al. (2025) to optimize our model. In addition, to prevent the optimized policy $\pi_\theta$ from diverging excessively from the reference model $\pi_{\text{ref}}$, a KL-divergence regularization term $\mathbb{D}_{\text{KL}}$ is introduced. The overall optimization objective is:

$$\max_{\pi_\theta} \mathbb{E}_{[q \sim D_{\text{rl}}, \{o_i\}_{i=1}^G \sim \pi_\theta(O|q)}$$
$$\left[ \frac{1}{G} \sum_{i=1}^G \min(\frac{\pi_\theta(o_i)}{\pi_{\theta_{\text{old}}}(o_i)} A_i, \text{clip}(\frac{\pi_\theta(o_i)}{\pi_{\theta_{\text{old}}}(o_i)}, 1 - \epsilon, 1 + \epsilon) A_i) - \beta\, \mathbb{D}_{\text{KL}}(\pi_\theta \| \pi_{\text{ref}}) \right], \tag{10}$$

where $\epsilon, \beta$ are the hyper-parameters.

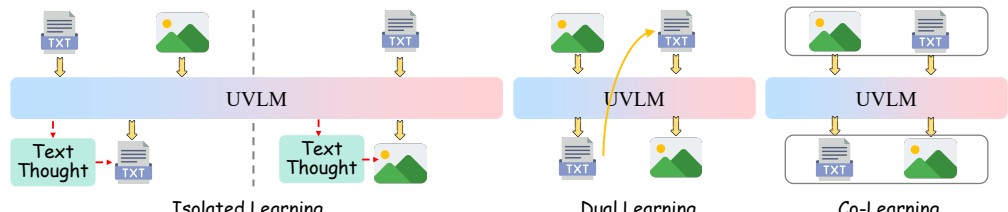

Figure 8: Comparison of existing mechanisms for synergizing understanding and generation, including isolated learning where the two abilities are trained independently, dual learning which leverages cross-modal reconstruction for mutual supervision, and co-learning which jointly optimizes both tasks with paired samples.

# E    COMPARISON WITH EXISTING SYNERGETIC LEARNING METHODS

We here position our approach in relation to existing methods for synergistic learning between understanding and generation. As illustrated in Fig. 8, prior work can be broadly grouped into two major directions.

**Unified Models for Synergistic Learning.**    A common line of research aims to unify understanding and generation within a single framework. Representative examples (Xie et al., 2025; Wu et al., 2025a; Deng et al., 2025; Xiao et al., 2025b) include models that adopt a purely autoregressive formulation, or hybrid designs that combine autoregressive and diffusion-based paradigms. In these approaches, synergy is encouraged through shared parameters, which enable implicit interaction between the two tasks. However, during training and inference, the two abilities remain largely independent. As a result, such models primarily learn to master understanding and generation in parallel, rather than achieving genuine synergy in task-solving.

**Learning Optimization for Synergistic Learning.**    Another approach is to design a learning schema that fosters synergy between understanding and generation. For instance, dual learning methods (Yan et al., 2025) translate visual inputs into textual descriptions and then regenerate visuals from those descriptions, with reconstruction quality serving as the optimization signal. Co-learning strategies (Jiang et al., 2025b) further extend this idea by coupling each sample with both generation prompts and multimodal understanding queries, thereby jointly improving performance on the two tasks. While effective for mutual supervision, these approaches still operate at the stage of co-training skills, rather than enabling active collaboration between them during task execution.

**Our Contribution.**    In contrast, our work introduces a fundamentally different perspective. We argue that real synergy should emerge not only during the learning phase but also in the problem-solving process itself. Specifically, we propose the Analyzing-Drafting problem-solving Loop (AD-Loop), a novel problem-solving mechanism in which understanding and generation are interleaved at appropriate moments to jointly address a task. Furthermore, we develop a two-stage training strategy that equips unified multimodal models with this capability. In particular, our second-stage hybrid learning scheme enables the model to adaptively and intelligently decide when to invoke understanding or generation, thereby achieving organic integration of the two abilities and realizing genuine synergy.

# F    DETAILED DATASET CONSTRUCTION

Here, we outline the process of constructing training data. The overall statistics are presented in Table 5, and an instance example visualization of each constructed dataset is shown in Fig. 9.

**AD-Loop Dataset for Understanding Task.**    To construct the interleaved analyzing–drafting loop for understanding tasks, we draw upon the following representative sources:

- **CMoT** (Cheng et al., 2025b) is a chain of multimodal-thought dataset that requires multimodal input and multi-output reasoning output. It consists of four categories: (1) Visual

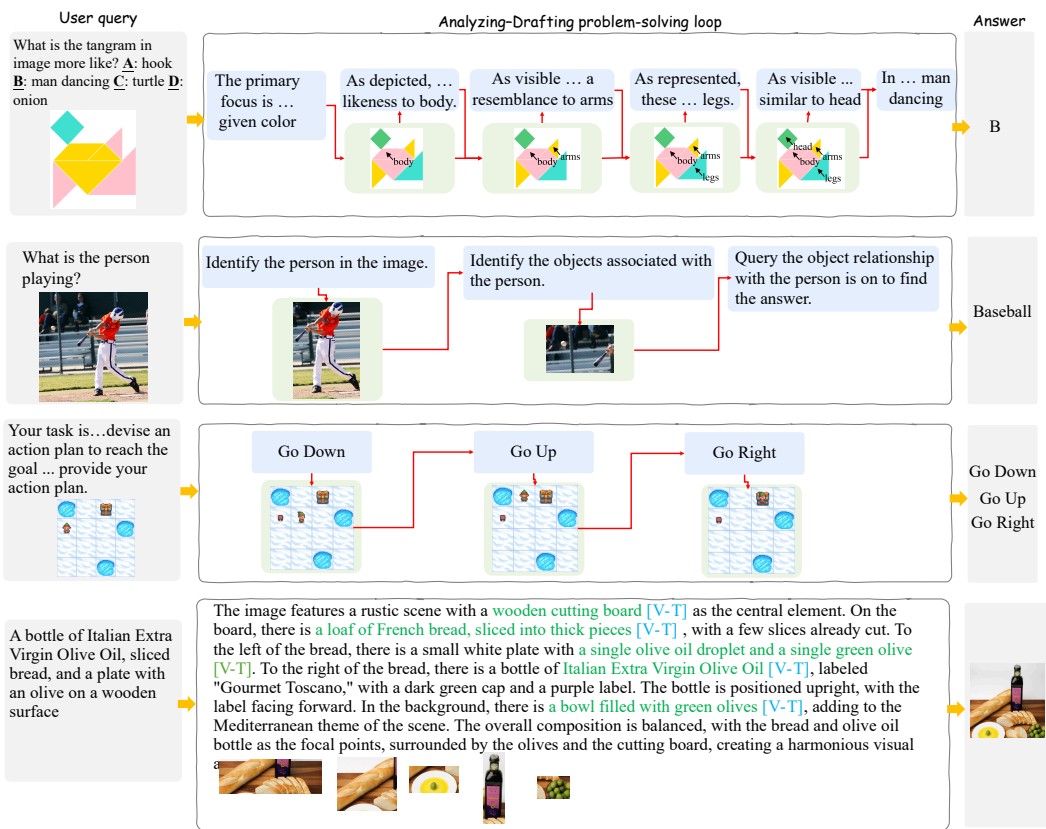

Figure 9: Visualization of the constructed dataset. Given a user query, we establish an analyzing–drafting problem-solving loop in which textual and visual thoughts alternate dynamically, ultimately yielding the final answer.

Creation, (2) Visual Deletion, (3) Visual Update, and (4) Visual Selection to comprehensively explore complex visual operations and concise expression in real scenarios. The final average number of AD-Loop is 3.

- **Visual-CoT** (Shao et al., 2024) provides intermediate reasoning annotations with bounding boxes that highlight critical regions necessary for answering visual reasoning questions. We derive explicit visual thoughts by zooming into and cropping the annotated regions, which serve as supervision signals for the visual-thought channel, resulting in an average of 2 per AD-Loop trajectory.
- **CoF** (Zhang et al., 2025b) identifies focus regions during question answering. We derive explicit visual thoughts by zooming into and cropping the annotated regions, which serve as supervision signals for the visual-thought channel, yielding an average of 3 per sample.
- Visual Spatial Planning (**VSP**) Yang et al. (2025b) formulates schematic grid-based navigation tasks, in which an agent must move from a designated start to a destination while avoiding "holes". Following Li et al. (2025), we render these states using the OpenAI Gym framework (Brockman et al., 2016), with the initial map and action sequence as inputs. Visual thoughts correspond to the visualization of each movement step, with an average of 3 latent visual thoughts.

**AD-Loop Dataset for Generation Task.** For generation tasks, we construct interleaved analyzing–drafting resources from the following datasets and models:

- **GoT-T2I** (Fang et al., 2025) provides both reasoning traces and bounding-box annotations of salient objects. We crop the corresponding image regions based on the bounding-box information and append these cropped patches directly after the associated textual descriptions, thereby forming explicit visual representations. We treat the annotated bounding boxes of salient objects as latent visual thoughts, resulting in an average of 4.

Table 5: The statistics of the constructed dataset for two-stage learning. 'Avg.' indicates the average number of latent visual thoughts within the datasets.

| Task | Dataset | Avg. | Stage-1 | Stage-2 | Total |
|------|---------|------|---------|---------|-------|
| Understanding | CMoT (Cheng et al., 2025b) | 3 | 1K | 1K | 2K |
| | Visual-CoT (Shao et al., 2024) | 2 | 12K | 1K | 13K |
| | CoF (Zhang et al., 2025b) | 3 | 4K | 1K | 5K |
| | VSP (Yang et al., 2025b) | 3 | 3K | 1K | 4K |
| Generation | GoT-T2I (Fang et al., 2025) | 4 | 12K | 3K | 15K |
| | Inter-T2I | 4 | 10K | 2K | 12K |

- **Inter-T2I** We leverage *X-to-Image* (Xiao et al., 2025a) corpora to obtain high-quality prompt–image pairs, and employ QVQ-72B-Preview (Team, 2024b) to construct reflective critiques conditioned on visual inputs. To enrich intermediate visual thoughts, we adopt *Flux1-dev* (Yang et al., 2024) to generate visual hypotheses guided by subgoals extracted from the prompts. We generate visual hypotheses based on sub-goals extracted from the prompts and retain only the first occurrence of each object to avoid duplication. The average number of latent visual thoughts is 4.

# G DETAILED SETTINGS

## G.1 TRAINING CONFIGURATION

We adopt two distinct UVLMs as the backbone of our framework. While both models incorporate dedicated heads for video understanding and generation, they differ fundamentally in their approaches to video synthesis: Janus-Pro employs an LLM backbone that generates *discrete tokens*, whereas BAGEL adopts a design that generates *continuous tokens*. Below, we detail the architectural designs of these two models.

BAGEL (7B) (Deng et al., 2025) is initialized from Qwen2.5 LLM (Qwen et al., 2025). For visual understanding, BAGEL adopts SigLIP2-so400m/14 (Tschannen et al., 2025) with a fixed 384-resolution to convert the raw pixels into tokens. For visual generation, BAGEL utilizes a pre-trained VAE model from FLUX (Yang et al., 2024) to convert images from pixel space to latent space and vice versa. The latent representation is then processed by a $2 \times 2$ patch embedding layer to reduce the spatial size and match the hidden dimension of the LLM backbone.

Janus-Pro (7B) (Chen et al., 2025c) utilizes the SigLIP (Zhai et al., 2023) to extract high-dimensional semantic features from images. These features are flattened from a 2-D grid into a 1-D sequence, and an understanding adaptor is used to map these image features into the input space of the LLM for understanding. For visual generation tasks, Janus-Pro adopts the VQ tokenizer from Sun et al. (2024), which discretizes images into token IDs for autoregressive generation.

During the training, the maximum number of each latent visual thought is set to 16 tokens. At stage 2, we set $\lambda = 1.0$ and a margin $\delta = 0.2$ for RL training. We set the weighting hyperparameters $\alpha = 1$ and $\gamma = 1$. We set the KL coefficient to $\beta = 0.001$ and $\epsilon = 0.5$. Besides the interleaved data, we also leverage 4K text-only reasoning data (Huang et al., 2025; Li et al., 2024) in Stage 2.

## G.2 EVALUATION METRICS

Here, we detail the evaluation metrics employed in the experiments. For understanding tasks, we report *accuracy*, following standard practice Xie et al. (2025); Deng et al. (2025); Ma et al. (2025), to measure the percentage of correctly predicted answers on each dataset. For generation tasks, we conduct evaluations on two benchmarks:

- **(1) GenEval**. We adopt the official GenEval toolkit[1] to assess text-to-image generation quality. GenEval comprises six task categories, including single object, two objects, counting, colors, position, and attribute binding. The GENEVAL *score* is formulated as a binary correctness measure, indicating whether all elements specified in the prompt are faithfully

---

[1]https://github.com/djghosh13/geneval

Table 6: Performance comparison of explicit and latent visual thoughts on the spatial visual planning task at levels 3 and 6.

| Type | Model | Level 3 | Level 6 |
|---|---|---|---|
| Explicit | Anole (Chern et al., 2025) | 0.02 | 0.00 |
| | MVoT (Li et al., 2025) | 0.21 | 0.03 |
| Implicit | Mirage (Yang et al., 2025b) | 0.75 | 0.39 |
| | Ours | **0.81** | **0.47** |

rendered in the generated image. This setup directly evaluates the model's compositional alignment between textual instructions and visual outputs.

- **(2) WISE**. We select a subset of WISE domains (e.g., biology, culture, and space) for evaluation. Following prior work Deng et al. (2025); Chen et al. (2025c), we report *WiScore*, the primary metric of the benchmark, which emphasizes the accuracy of depicted objects and entities grounded in world knowledge. WiScore is computed as a weighted combination of three components: Consistency, Realism, and Aesthetic Quality. Higher WiScore values indicate stronger capability in generating images that correctly represent real-world concepts while maintaining visual plausibility.

## H    EXTENDED EXPERIMENTAL RESULTS

**Comparison of Explicit and Implicit Visual Thoughts.**    We compare existing explicit and implicit approaches on the spatial visual planning task. As shown in Table 6, explicit methods yield the weakest performance. A likely reason is that when the model is required to explicitly draft intermediate visual thoughts, the quality of these drafts is heavily constrained by the model's inherent generation capability. In such cases, the model struggles to construct faithful visualizations from high-level semantic representations, resulting in low-quality visual thoughts. Building upon these imperfect drafts further propagates errors, resulting in cumulative degradation of reasoning. By contrast, latent (implicit) visual thoughts achieve consistently stronger results, as they bypass the limitations of explicit drafting and enable more faithful internal reasoning representations.

**Visualization on Understanding Tasks.**    We further provide qualitative visualizations of the model's performance on various understanding tasks. For instance, in mathematical reasoning problems (Fig. 11), the model is first able to analyze the question, then drafts intermediate visual sketches analogous to human scratch work, and finally derives the correct solution. Similarly, in more abstract scenarios, such as tangram analysis (Fig. 12), the model successfully generates accurate intermediate visualizations that facilitate the correct interpretation, significantly outperforming approaches like BAGEL, which also employ self-thinking strategies.

**Visualization on Generation Tasks.**    We also provide additional qualitative results on generation tasks. Compared with alternative reasoning-based methods, our model demonstrates superior performance on prompts that require commonsense reasoning. For example, as shown in Fig. 13, baseline models tend to generate kangaroos sitting directly on the ground, whereas our model correctly depicts the kangaroo using its tail for support, which is more consistent with reality. In creative synthesis scenarios, such as the counterfactual prompt "a plant watering a gardener" our model is still able to produce outputs that faithfully capture the user's intent. Furthermore, Fig. 14 contrasts generations produced by the base model with and without our thinking mechanism, where our approach consistently yields higher-quality images that better align with user instructions.

**Visualization on Editing Tasks.**    We further compare our approach with BAGEL (Deng et al., 2025) on diverse image editing tasks, both with and without the "think" process, as shown in Fig. 15. The results indicate that our method consistently produces more faithful and semantically aligned edits. For instance, when asked to infer the missing element in a visual pattern, BAGEL either fails to capture the correct reasoning or generates unrelated content, whereas our model accurately identifies the missing orange. In procedural editing tasks such as heating corn kernels until they pop, our approach generates realistic popcorn, while BAGEL outputs less plausible textures. Similarly,

for biological transformations (e.g., albinism in corals) and structural corrections (e.g., fixing unreasonable parts of a bicycle), our method yields coherent results that respect both semantic intent and visual fidelity. These comparisons highlight the strength of our interleaved thinking process in producing edits that are both accurate and visually credible.

**Challenging Cases.** Fig. 16 and 17 present several challenging examples where our model still exhibits limitations. On the understanding side, the model may occasionally over-draft on simple queries. For very basic attribute questions, the introduction of additional reasoning steps can lead to unnecessary elaboration and, in some cases, incorrect predictions. Similarly, although the model shows improvements on visually grounded mathematical tasks, such as MathVista (Lu et al., 2023), it still struggles on problems that require strict symbolic reasoning, where applying AD-Loop may introduce deviations from the optimal logical path.

On the generation side, the model continues to face difficulties in rendering fine-grained visual details, including scene text, subtle body parts (e.g., fingers), and numerically precise elements. A plausible explanation is that the compact latent visual representations may under-encode such fine details. We anticipate that incorporating more detailed feedback into the Stage-2 RL reward, encouraging the model to attend to fine-grained latent visual thoughts, could further mitigate these issues in future work.

## I   THINKING STRATEGIES COMPARISON WITH EXISTING VLMS VIA VISUAL INFORMATION

Compared to existing "alternation-based" or "integration-based" visual information argumented multimodal reasoning works (Zhang et al., 2025a; Zheng et al., 2025), the proposed AD-Loop is fundamentally different across several objective and design perspectives.

**(1) Objective: achieving genuine synergy between understanding and generation**. The goal of AD-Loop is not merely to alternate modalities but to enable a mutual strengthening between the model's understanding and generation capabilities. The entire framework, both in design and training, is centered around this objective, which distinguishes it from prior works that only combine modalities without enabling reciprocal influence.

**(2) Modeling perspective: internal, bidirectional, and iterative interaction rather than passive alternation**. In AD-Loop, both textual and visual information are derived from the model's internal reasoning without relying on external tools for visual integration. The process is not a fixed alternation but a dynamic switching between analytic (textual) and synthetic (visual-drafting) modes, allowing iterative refinement of reasoning. Moreover, unlike prior methods that use explicit images or pixel-level drafts, AD-Loop introduces latent visual thoughts, which serve as concept-aligned abstractions. These representations compress visual regions into semantic clusters and disentangle relevant from irrelevant information, enabling visual reasoning without pixel-level noise or computational overhead.

**(3) Learning perspective: adaptive rather than mandatory or heuristic alternation.** As discussed in Appendix E, existing synergistic or alternating approaches typically rely on fixed schedules or heuristics. In contrast, AD-Loop is explicitly adaptive: our hybrid learning scheme with Stage-2 group-relative RL allows the model to intelligently decide when to invoke understanding or drafting. This results in an organic integration of the two capabilities and is key to realizing genuine synergy, rather than superficial alternation.

[**Question**]: were any of the objects in the initial frame that you can still see in the second frame moved from their original positions?
[**Correct answer**]: chair was moved left and towards from the camera

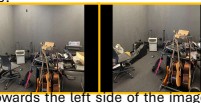

[**BAGEL w/ Think**] To determine if any objects were moved from their original positions between the two frames, we need to carefully compare the positions of the objects in both images.\n\n1. **Identify Key Objects:**\n  - **Chair:** The chair is a prominent object in both frames.\n  - **Guitars:** The guitars are also noticeable in both frames.\n  - **Desk:** The desk and its contents are another key element.\n\n2. **Compare Positions:**\n  - **Chair:** In the first frame, the chair is positioned towards the left side of the image, closer to the camera. In the second frame, the chair is moved to the right side of the image, further away from the camera.\n - **Guitars:** The guitars appear to be in the same general area in both frames, but their exact positions might have changed slightly.\n  - **Desk:** The desk and its contents seem to have shifted slightly, but the main items (like the box and the computer tower) are still in the same general area.\n\n3. **Conclusion:**\n  - The chair has been moved from its original position. It was moved to the right and away from the camera.\n\nTherefore, the correct answer is: 'chair was moved right and away from the camera'.\n\nFinal answer: **'chair was moved right and away from the camera'**

[**Janus-Pro**] Step 1: Observe the initial frame and the second frame.\n- In the initial frame, there is a chair on the left side of the room.\n- In the second frame, the chair is still on the left side but slightly closer to the camera.\n\nStep 2: Analyze the movement of the objects.\n- The chair appears to have moved slightly towards the camera, but it is still in the same general position relative to the other objects.\n\nStep 3: Compare the options.\n- Option 1 suggests the chair was moved left and towards the camera.\n- Option 2 suggests the chair was moved right and away from the camera.\n\nStep 4: Determine the correct answer.\n- The chair's position in the second frame does not match the description of moving left or right and towards or away from the camera.\n\nFinal answer: **chair was not moved from its original position**.

[**Harmon**] To determine if any objects were moved, we need to compare the positions of the items in both frames. In the initial frame (left side of the image), there is a chair that appears to be facing the camera and has its legs spread out for stability. In the second frame (right side of the image), the same chair seems to have been repositioned with its legs now closer together and angled slightly away from the camera.\n\nIn the second frame, the chair's position relative to the camera has changed: it was initially on the left but is now more centered in the frame. This indicates that the chair was indeed moved from its original position.\n\nTherefore, the correct answer is:\n\n[**'chair was moved right and away from the camera'**]

[**Ours**] To decide whether an object moved between frames, we compare relative positions of key items. In the initial Frame 1 the rolling chair sits beside the guitar case on the floor [V-T], with several guitars nearby and the table close to its right.\n In the Frame 2, only part of the chair is visible and it appears behind the guitar case [V-T], indicating a shift toward the camera and leftward relative to the original layout; it's also farther from the table than before.\n Final answer:\n[**chair was moved left and towards from the camera'**]

Figure 10: Qualitative comparison across baseline unified MLLMs on multimodal understanding. Correct answers are marked in green, errors in red, and [V-T] denotes latent visual thoughts.

---

[**Question**]: AB is the diameter of ⊙O. Point C is a point on AB. Point D is on ⊙O, AD = AC. Connect DC and extend it to intersect ⊙O at point E. Connect AD, AE, OE. If ∠BAD = 30°, the degree measure of ∠COE is ()
[**Option**]: A: 30°    B: 35°    C: 40°    D: 45°
[**Correct Answer**]: A

[**BAGEL w/o Think**] **Final answer: B**

[**BAGEL w/ Think**] To find the measure of ∠COE, we can use properties of circles and angles. Let's break down the problem step by step.\n\n1. **Identify the given information:**\n  - AB is the diameter of circle O.\n  - Point C is on AB.\n  - Point D is on circle O such that AD = AC.\n  - AD = AC implies that triangle ADC is isosceles with AD = AC.\n  - ∠BAD = 30°.\n\n2. **Understand the properties of circles:**\n  - The angle subtended by a diameter in a circle is a right angle (90°). Therefore, ∠ADB = 90°.\n  - Since AD = AC, triangle ADC is isosceles, so ∠ADC = ∠ACD.\n\n3. **Calculate ∠ADC:**\n  - Since ∠BAD = 30° and ∠ADB = 90°, ∠ABD = 60° (since the angles in a triangle sum to 180°).\n  - In triangle ADC, ∠ADC = ∠ACD. Let's denote each of these angles as x.\n  - Therefore, 2x + 30° = 180° (since the sum of angles in triangle ADC is 180°).\n  - Solving for x: 2x = 150°, so x = 75°.\n  - Thus, ∠ADC = 75°.\n\n4. **Determine ∠COE:**\n  - Since ∠ADC = 75°, and ∠ADC is an inscribed angle that subtends arc AC, the measure of arc AC is 2 * 75° = 150°.\n  - The central angle ∠COE that subtends the same arc AC is equal to the measure of the arc, which is 150°.\n  - However, ∠COE is the angle at the center of the circle, and since ∠COE is the central angle for arc AC, it is equal to the measure of the arc AC, which is 150°.\n\n5. **Calculate the remaining angle:**\n  - Since AB is the diameter, the total angle around point O is 360°.\n  - The angle ∠AOE is the remaining angle that completes the circle.\n  - ∠AOE = 360° - 150° = 210°.\n\n6. **Determine the correct option:**\n  - The question asks for the measure of ∠COE, which we found to be 150°.\n  - However, the options provided are A. 30°, B. 35°, C. 40°, D. 45°.\n  - None of the options match 150°, which suggests there might be a misunderstanding in the problem setup or options provided. However, based on the given options and the logical steps, the closest and most reasonable answer would be D. 45°, as it is the only option that could be a part of the correct calculation process.\n\n**Final answer: D.**

[**Janus-Pro**] Step 1: Identify the key elements in the problem.\n- ⊙O is the circle.\n- AB is the diameter of ⊙O.\n- Point C is a point on AB.\n- Point D is on ⊙O, AD = AC.\n- Connect DC and extend it to intersect ⊙O at point E.\n- Connect AD, AE, OE.\n- ∠BAD = 30°.\n\nStep 2: Analyze the given information.\n- Since AD = AC, triangle ACD is isosceles.\n- ∠BAD = 30°, so ∠CAD = 60°.\n- Since AD = AC, ∠ADE = 60°.\n- ∠ADE + ∠COE = 180°.\n\nStep 3: Calculate ∠COE.\n- ∠ADE = 60°.\n- ∠ADE + ∠COE = 180°.\n- 60° + ∠COE = 180°.\n- ∠COE = 180° - 60°.\n- ∠COE = 120°.\n\nStep 4: Determine the correct answer.\n- The degree measure of ∠COE is 120°.\n\n**Final answer: 120°**

[**Ours**] <think>To answer this question, the image is depicted below: [V-T]\n Let's think step by step: 1) Let O(0,0), A(-1,0), B(1,0). Then AB is the diameter. 2) ∠BAD = 30° ⇒ arc BD = 60°, so D = (1/2, √3/2). 3) Triangle ABD is a 30°–60°–90° right triangle, hence AD = √3. Given AC = AD and C lies on AB, we get C = (√3–1, 0). 4) Line DC intersects the circle again at E = (√3/2, –1/2). 5) Vector OC lies along the x-axis (0°), OE has angle –30°. Therefore, ∠COE = 30°. </think><answer>**Final answer: A**</answer>

Figure 11: Understanding Comparison across UVLMs on the math question answering task. Correct answers are marked in green, errors in red, and [V-T] denotes latent visual thoughts.

[**Question**]: What does the tangram in the image show a closer likeness to?
[**Option**]: A: a dog's head      B: someone flexin      C: mailbox      D: woman
[**Correct Answer**]: C

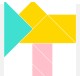

[**BAGEL w/o think**] Final answer: B

[**BAGEL w/ think**] To determine what the tangram in the image resembles, we need to carefully analyze the shapes and their arrangement. \n\n1. **Identify the Shapes**: The tangram consists of seven pieces: five triangles, one square, and one parallelogram. These pieces are arranged to form a larger shape.\n\n2. **Analyze the Arrangement**:\n   - The largest triangle is positioned on the left side.\n   - The square is located at the top center.\n   - The parallelogram is at the bottom center.\n   - The remaining triangles are arranged around these central shapes.\n\n3. **Compare with Options**:\n   - **Option A (a dog's head)**: The arrangement does not resemble a dog's head. The shapes do not form the typical features of a dog's face.\n   - **Option B (someone flexin)**: The shapes do not form a person flexing their muscles. The arrangement lacks the typical features of a human figure.\n   - **Option C (mailbox)**: The arrangement does not resemble a mailbox. The shapes do not form the typical rectangular and triangular features of a mailbox.\n   - **Option D (woman)**: The arrangement of the shapes does resemble a woman. The largest triangle can be seen as the head, the square as the upper body, and the parallelogram as the lower body. The other triangles can be interpreted as arms and legs.\n\n4. **Conclusion**: Based on the analysis, the tangram in the image most closely resembles a woman.\n\n**Final answer: D.**

[**Janus-Pro**] To answer this question, let's analyze the tangram shapes and their arrangement in the image.\n\n1. The image shows a set of geometric shapes that are typically used in tangram puzzles.\n2. The shapes are arranged in a way that resembles a dog's head.\n3. The yellow shapes form the outline of the dog's head, with the pink shape representing the body.\n4. The cyan shape is positioned to resemble the ears or side of the dog.\n\nBased on this analysis, the tangram in the image shows a closer likeness to a dog's head.\n\n**Final answer: A**

[**MindOmni**] To determine what the tangram in the image shows a closer likeness to, let's analyze the shapes and their arrangement:\n\n1. The image consists of several geometric shapes: triangles, a square, and a parallelogram.\n2. The shapes are arranged in a way that suggests a recognizable form.\n3. The large yellow triangle at the top left resembles an ear.\n4. The two yellow squares and the yellow triangle below them form a shape that looks like a face with a mouth.\n5. The pink shapes at the bottom resemble a body.\n\nConsidering these observations, the overall arrangement of the shapes most closely resembles a face with a mouth, which is characteristic of a dog's head.\n\n**Final answer: A. a dog's head**

[**Ours**] <think>The initial step involves examining regions of the image that are marked by a specific color.\nAs illustrated, these regions resemble the shape of a pole **[V-T]**.\nNext, attention shifts to segments highlighted by another color. As shown, these areas take on a form similar to a box **[V-T]**.\nFinally, we focus on the portions rendered in the remaining colors.\nThese sections, as depicted, bear a likeness to a flag **[V-T]** .\n Taken together, the overall tangram arrangement most closely aligns with the shape of a mailbox **[V-T]** .\n </think> \n\n <answer> **Final answer: C**</answer>

Figure 12: Understanding Comparison across UVLMs on multimodal content reasoning task. Correct answers are marked in green, errors in red, and [**V-T**] denotes latent visual thoughts.

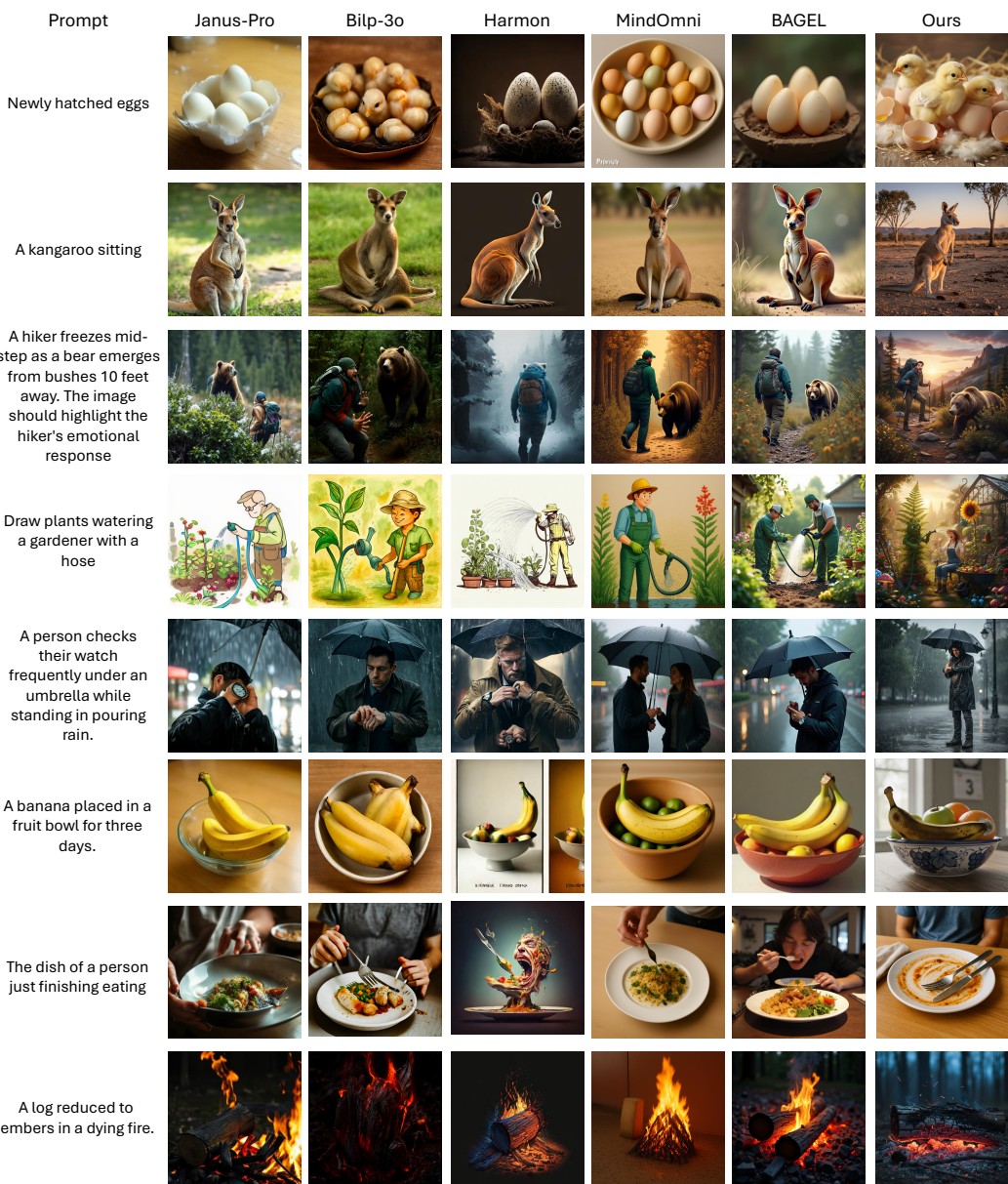

Figure 13: Generation Comparison across UVLMs.

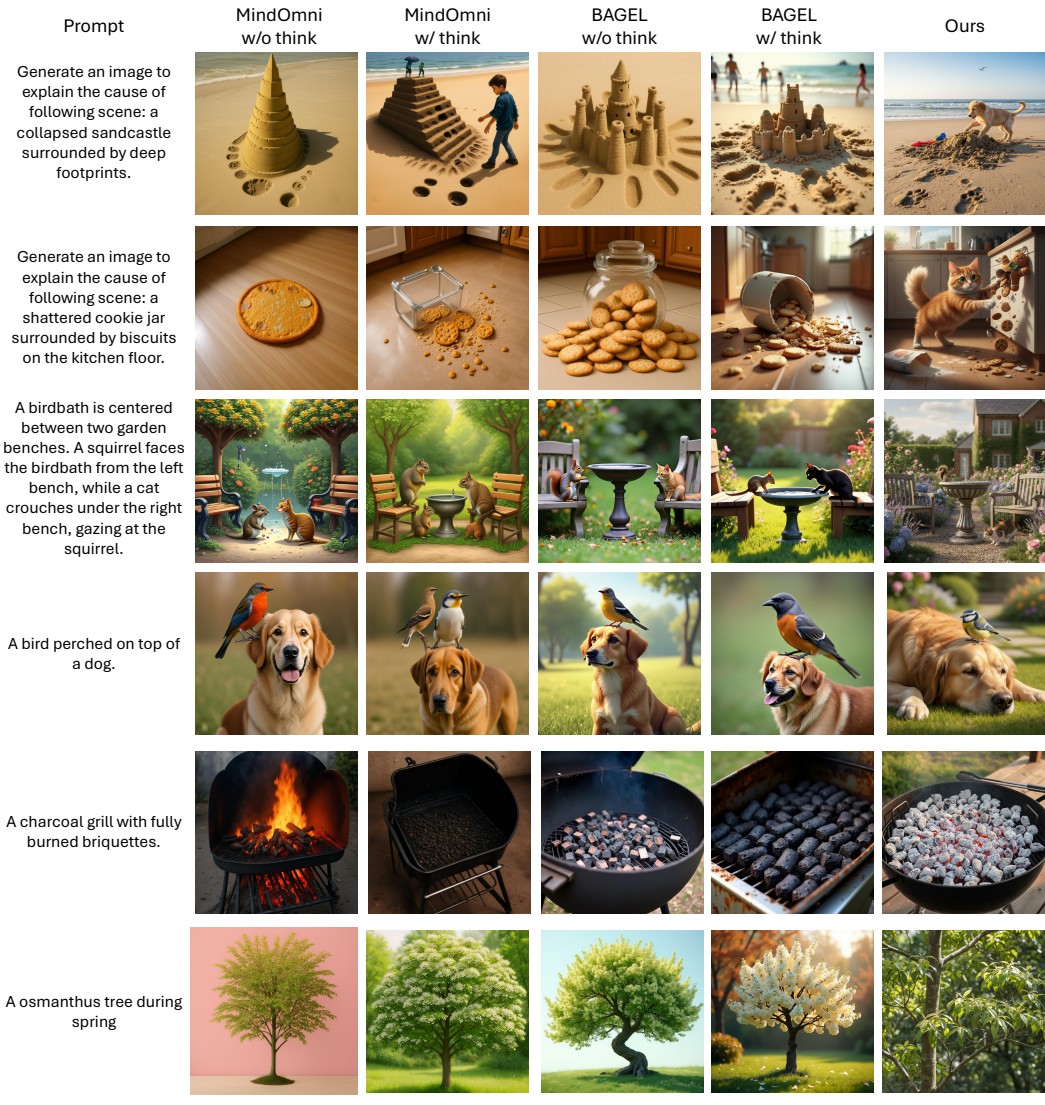

Figure 14: Generation Comparison on MindOmni (Xiao et al., 2025b) and BAGEL (Deng et al., 2025) with/without Think process.Our proposed method achieves more accurate generation results.

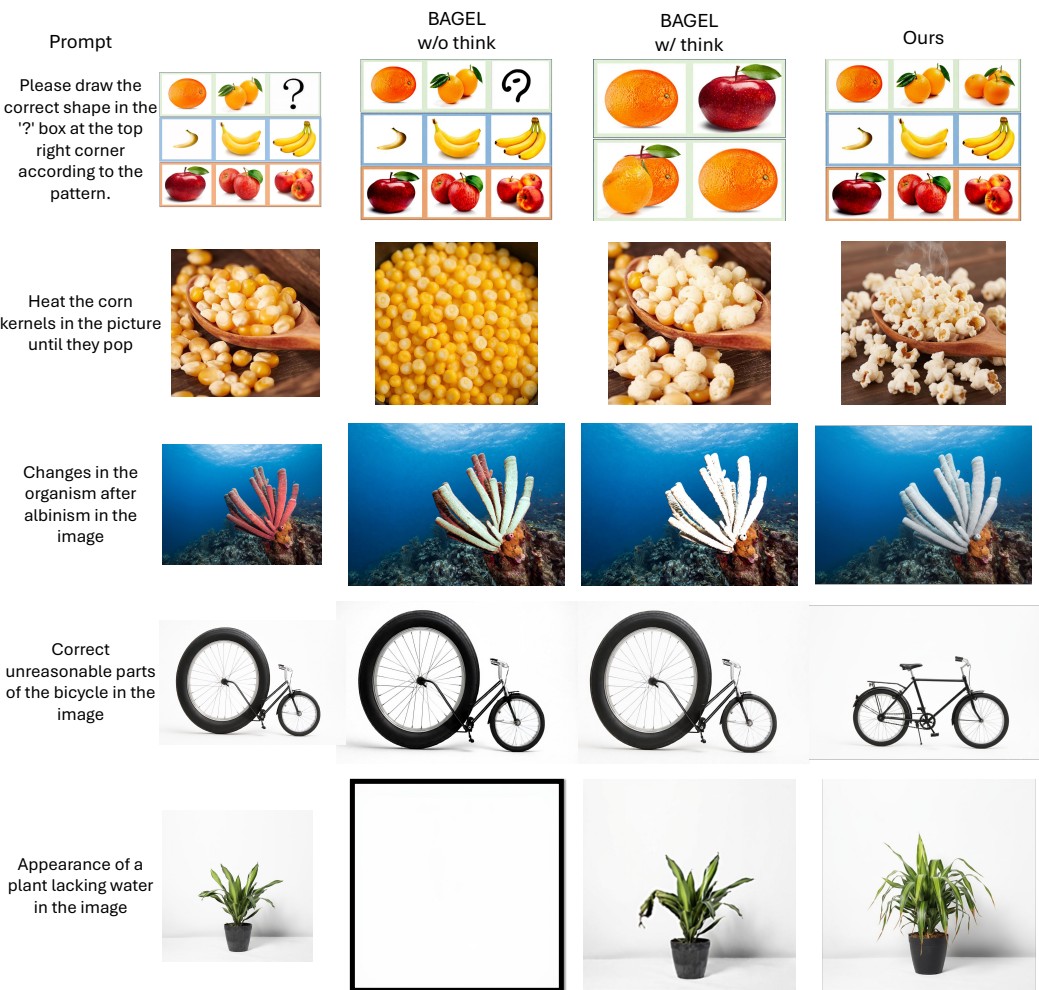

Figure 15: Editing Comparison on BAGEL (Deng et al., 2025) with/without think process.

**[Question]** What is the most prominent feature in the image?
**[Option]** A: The skyline, B: The golf course, C: The trees, D: The person

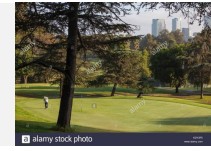

**[BAGEL w/o Think]** Final answer: C.

**[Ours]** The image is taken from a perspective that places the golf course in the foreground **[V-T]**, with the trees and skyline serving as secondary elements **[V-T]**. The golf course is the central focus, with the green and the golfer being the main subjects **[V-T]** . The trees frame the scene and add depth, but they are not the primary focus **[V-T]** . The skyline, while visible, is in the background and does not dominate the image **[V-T]** . Therefore, the most prominent feature is the golf course. Final answer: B.

- - - - - - - - - - - - - - - - - - - - - - - - - - - - - - - - - - - - - - - - - - - - - - - - - - -

**[Question]** Find the greatest common divisor of $3339$, $2961$, and $1491$.

**[BAGEL w/ Think]** Let's compute it step-by-step using the Euclidean algorithm.\n\n1. Find gcd(3339, 2961)\n3339 − 2961 = 378\nSo now compute gcd(2961, 378):\n2961 ÷ 378 = 7 × 378 = 2646\n2961 − 2646 = 315\nNow gcd(378, 315):\n378 − 315 = 63\ngcd(315, 63) = 63\nSo, gcd(3339, 2961) = 63\n\n2. Now compute gcd(63, 1491)\n1491 ÷ 63 = 23 × 63 = 1449\n1491 − 1449 = 42\nNow gcd(63, 42):\n63 − 42 = 21\nSo, gcd(63, 1491) = 21\n\nFinal Answer:\n21

**[Ours]** To find the greatest common divisor, we first determine the greatest common divisor of the first two numbers: $\gcd(3339, 2961)$. \n$3339 = 1 \cdot 2961 + 378$\n$2961 = 7 \cdot 378 + 318$\n$378 = 1 \cdot 318 + 60$\n$318 = 5 \cdot 60 + 18$\n$60 = 3 \cdot 18 + 6$\n$18 = 3 \cdot 6 + 0$\nTherefore, $\gcd(3339, 2961) = 6$. \n **[V-T]** Next, we find the greatest common divisor of the result from the previous step (6) **[V-T]** and the third number (1491): $\gcd(6, 1491)$ . \n$1491 = 248 \cdot 6 + 3$\n$6 = 2 \cdot 3 + 0$\n **[V-T]** Therefore, $\gcd(6, 1491) = 3$. \nThus, the greatest common divisor of all three numbers is $\mathbf{3}$.\nFinal answer: 3.

Figure 16: Failure cases illustrating over-drafting on simple tasks, as well as challenges on problems that require strictly logical mathematical reasoning.

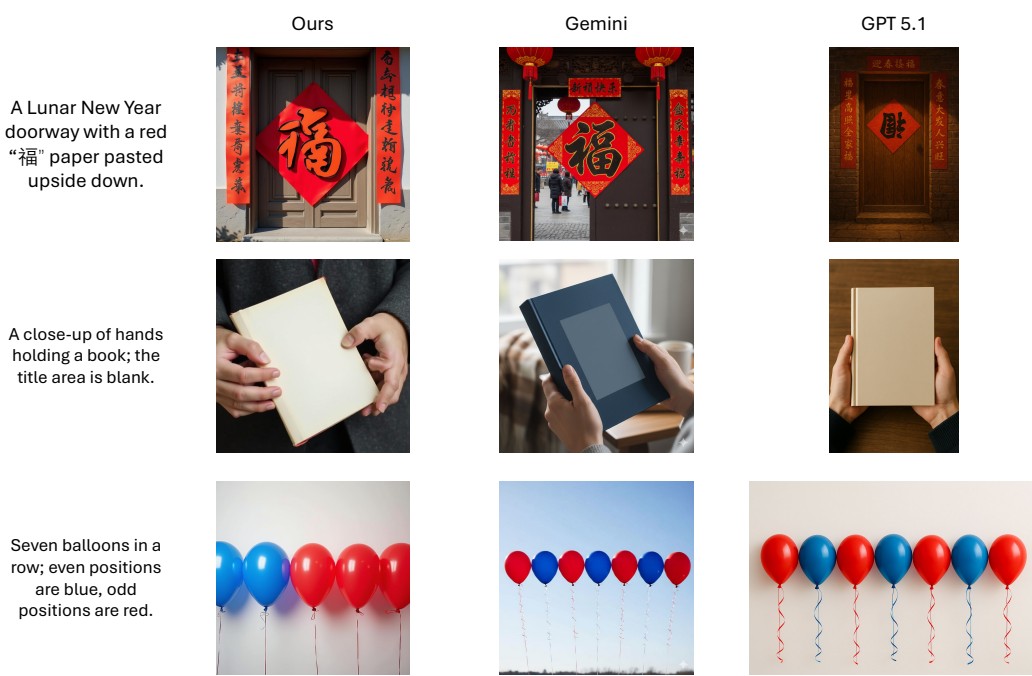

Figure 17: Failure cases showing that the proposed method still struggles with rendering scene text, as well as fine-grained body details and numerically precise content.

