# OpenReview forum: "Synergizing Understanding and Generation with Interleaved Analyzing-Drafting Thinking"
_ICLR.cc/2026/Conference — ICLR 2026 Poster_

### Official Review · Reviewer_tVfz · 2025-10-31

**Soundness:** 3
**Presentation:** 3
**Contribution:** 3
**Rating:** 8
**Confidence:** 3

**Summary:**

This paper proposes a new thinking paradigm called the Alternating Analysis–Drafting Problem-Solving Loop (AD-Loop), designed to achieve genuine synergy between the understanding and generation capabilities in Unified Vision-Language Models (UVLMs). The AD-Loop enables the model to iteratively refine its comprehension and output by dynamically alternating between textual thoughts (for analysis and reasoning) and visual thoughts (for sketching and spatial layout).

**Strengths:**

The AD‑Loop paradigm is proposed, emphasizing explicit, dynamic, and reciprocal interaction between understanding and generation. This concept goes beyond mere architectural unification, offering a new approach for achieving deeper levels of general multimodal intelligence.

**Weaknesses:**

Existing MLLM research has begun to explore the alternation or integration of textual and visual information. This paper needs to more clearly define the fundamental differences between AD-Loop and such works — specifically, how AD-Loop achieves genuine synergy rather than merely alternation.

**Questions:**

Please elaborate on the specific composition and weight distribution of the internal reward (Intra-reward) and external reward (Inter-reward) in S2. In particular, how is the internal reward designed to encourage “intelligent and autonomous” alternating decision-making?

---

> ### Author Response · Authors · 2025-11-22
>
> We sincerely thank Reviewer #tVfz for the positive evaluation and highly insightful questions.
> Below, we provide detailed clarifications addressing each point.
>
> -----
>
> **Q1.Existing MLLM research has begun to explore the alternation or integration of textual and visual information. This paper needs to more clearly define the fundamental differences between AD-Loop and such works, specifically, how AD-Loop achieves genuine synergy rather than merely alternation.**
> **A**: We clarify that AD-Loop is fundamentally different from existing works across several objective and design perspectives.
>
> *(1) Objective: achieving genuine synergy between understanding and generation*.
> The goal of AD-Loop is not merely to alternate modalities but to enable a mutual strengthening between the model’s understanding and generation capabilities. The entire framework, both in design and training, is centered around this objective, which distinguishes it from prior works that only combine modalities without enabling reciprocal influence.
>
> *(2) Modeling perspective: internal, bidirectional, and iterative interaction rather than passive alternation*.
> In AD-Loop, both textual and visual information are derived from the model’s internal reasoning without relying on external tools for visual integration. The process is not a fixed alternation but a dynamic switching between analytic (textual) and synthetic (visual-drafting) modes, allowing iterative refinement of reasoning. Moreover, unlike prior methods that use explicit images or pixel-level drafts, AD-Loop introduces latent visual thoughts, which serve as concept-aligned abstractions. These representations compress visual regions into semantic clusters and disentangle relevant from irrelevant information, enabling visual reasoning without pixel-level noise or computational overhead.
>
> *(3) Learning perspective: adaptive rather than mandatory or heuristic alternation*.
> As discussed in Appendix E, existing synergistic or alternating approaches typically rely on fixed schedules or heuristics. In contrast, AD-Loop is explicitly adaptive: our hybrid learning scheme with Stage-2 group-relative RL allows the model to intelligently decide when to invoke understanding or drafting. This results in an organic integration of the two capabilities and is key to realizing genuine synergy, rather than superficial alternation.
>
> We add discussions in the revised version to clearly highlight these distinctions.
>
>
>
>
> ------
> **Q2.Please elaborate on the specific composition and weight distribution of the internal reward (Intra-reward) and external reward (Inter-reward) in S2. In particular, how is the internal reward designed to encourage “intelligent and autonomous” alternating decision-making?**
> **A**:
> Thank you for raising this question. This question touches the core of Stage-2 training.
> We elaborate the reward design more clearly below.
>
> For each query, we sample two groups of responses: a $\\mathcal{V}+$ group with AD-Loop enabled and a $\\mathcal{V}-$ group using text-only reasoning. The rewards serve two complementary purposes.
>
> - *Internal reward (Intra-reward)* is computed within each group to assess the quality of responses under the same reasoning mode. It enables the model to learn which instances of AD-Loop thinking are genuinely effective and which cases can be adequately handled by text-only reasoning. To avoid spurious improvements, we apply a margin-based bonus so that AD-Loop does not receive credit unless it yields a meaningful gain, thereby preventing unnecessary usage.
>
> - *External reward (Inter-reward)* compares the best trajectory between the two groups and identifies whether AD-Loop or text-only reasoning performs better for that specific query. It therefore learns the preference between the two modes, encouraging the model to choose AD-Loop only when it provides a clear benefit, rather than defaulting to it.
>
> The combined advantage results in a learned meta-policy of ``When do I need AD-Loop thoughts, and when is text enough?``, which is precisely what enables AD-Loop to achieve intelligent and autonomous alternation rather than fixed or heuristic switching.

---

### Official Review · Reviewer_PeBS · 2025-10-31

**Soundness:** 4
**Presentation:** 3
**Contribution:** 3
**Rating:** 8
**Confidence:** 2

**Summary:**

The authors propose a new method to improve the capabilities of unified vision-language models. They propose a new approach that combines analytic processes and drafting thoughts (textual and visual thoughts).  They design a two stage strategy for training consisting of supervised learning and RL.  They show performance improvements for benchmark tests for understanding and generation.

**Strengths:**

The authors address an important issue with what seems to be an innovative approach.

The comprehensive evaluation for understanding and generation is also a strength.

Ablation of the thinking types section was also a strength.

Good discussion addressing interesting questions spanning extensions into other MLLMs,  whether visual thoughts should be derived from understanding vs. the generation encoder, and the visualization of implicit visual thoughts.

Very thorough methods section with detailed descriptions.

**Weaknesses:**

I was unclear about how novel the work is. There are many publications using visual representations (imagination) to augment language reasoning. Please expand on how this is different.

Please show some examples in Figure 7 where the proposed method did not generate better results. This would also be interesting.

There were parts where I had a difficult time following the methods and even the evaluation.  The paper is dense, but other reviewers more expert in the field might have an easier time following the paper.

**Questions:**

see weaknesses

---

> ### Author Response · Authors · 2025-11-22
>
> We sincerely thank Reviewer PeBS for the positive assessment and the helpful suggestions.
> Below, we address each of your points with detailed clarifications and additional examples.
>
> ------
> **Q1.I was unclear about how novel the work is. There are many publications using visual representations (imagination) to augment language reasoning. Please expand on how this is different.**
> **A**: Thank you for raising this important question.
> We agree that several prior works incorporate visual information into reasoning, but AD-Loop differs from them in three fundamental ways:
>
> *(1) Unified use of visual representations for both understanding and generation*.
> Existing approaches typically leverage visual signals for either visual understanding or for guiding generation, but not both within a single reasoning framework. To the best of our knowledge, AD-Loop is the first to employ visual representations in a unified manner that simultaneously benefits both modalities.
>
> *(2) Bidirectional and iterative synergy rather than unidirectional augmentation*.
> In AD-Loop, visual drafts are modeled as part of the model’s generative capability, while textual reasoning reflects its understanding capability. The framework enables the model to iteratively transition between analyzing and drafting, allowing the two abilities (i.e., understanding and generation) to complement and refine one another in a problem-solving loop.
>
> *(3) Explicit adaptiveness instead of fixed or mandatory alternation*.
> Prior works either always invoke visual imagination or rely on heuristically triggered usage. In contrast, AD-Loop learns an adaptive gating mechanism through group-relative RL, allowing the model to automatically decide when visual thoughts are beneficial. As a result, visual drafts become active reasoning components rather than auxiliary cues.
>
>
>
>
>
>
> ------
> **Q2.Please show some examples in Figure 7 where the proposed method did not generate better results. This would also be interesting.**
> **A**:
> Thank you for raising this important question.  Concerning the generation side, we show some failure cases in Appendix Figure 17 where the model continues to face difficulties in rendering fine-grained visual details, including scene text, subtle body parts (e.g., fingers), and numerically precise elements. A reasonable explanation is that the compact latent visual representations may under-encode such fine details. We anticipate that incorporating more detailed feedback into the Stage-2 RL reward, encouraging the model to attend to fine-grained latent visual thoughts, could further mitigate these issues in future work.
>
>
>
>
> ------
> **Q3.There were parts where I had a difficult time following the methods and even the evaluation. The paper is dense, but other reviewers more expert in the field might have an easier time following the paper.**
> **A**: Thank you for pointing this out, we agree that the original submission was dense due to space constraints. To improve clarity, we have refined both the method description and evaluation presentation in the revised version.
>
> First, we streamlined the methodological overview. Our approach builds on unified multimodal models such as BAGEL and Janus-Pro (details provided in Appendix G.1). The proposed method consists of two core components:
>
>     - Reasoning mechanism: Section 3.1 introduces the AD-Loop process, where textual and latent visual thoughts are interleaved during reasoning.
>
>     - Learning strategy: Section 3.2 presents the supervised Stage-1 training using initially constructed AD-Loop traces, and Section 3.3 describes the Stage-2 reinforcement learning that enables the model to adaptively decide when to perform analyzing or drafting.
>
> Second, to improve the readability of the evaluation, we added clearer descriptions of the metrics and benchmark settings in Appendix G.2.
>
> We hope these revisions make the paper substantially easier to follow.

---

### Official Review · Reviewer_h8QD · 2025-10-31

**Soundness:** 3
**Presentation:** 3
**Contribution:** 3
**Rating:** 4
**Confidence:** 3

**Summary:**

The paper introduces the Interleaved Analyzing–Drafting Loop (AD-Loop), a framework for unified vision-language models that alternates between textual analysis and visual drafting during reasoning. Through supervised fine-tuning on interleaved reasoning traces and reinforcement learning using Group-Relative Preference Optimization (GRPO), the model learns to decide when to analyze or draft. Applied to UVLMs (BAGEL-7B), the method improves both visual understanding and image generation, setting new state-of-the-art results on several benchmarks.

**Strengths:**

1. The interleaved analyzing–drafting mechanism establishes a tighter synergy between vision understanding and generation than prior “unified” models. This new paradigm addresses a clear gap by turning generation and analysis into mutually reinforcing steps rather than independent skills.

2. The two-stage training strategy (SFT of interleaved reasoning followed by RL) is well-motivated. This pipeline enables the model to learn the complex analyze-then-draft procedure in a guided way, then optimize adaptively via reinforcement learning. The use of preference-based rewards and an autonomous switching policy shows a way to balance the dual objectives.

3. AD-Loop yields consistent improvements across diverse benchmarks for both modalities. The approach outperforms baselines on multiple datasets and even when applied to different underlying model architectures, indicating broad applicability.

**Weaknesses:**

1. The framework introduces significant complexity in both training and inference. It requires a specialized two-phase training (including an RL stage), and at runtime the model must perform multiple analyze–draft iterations per query. This likely incurs substantial computational cost and latency compared to standard one-pass models. The paper does not discuss inference speed or resource requirements, which raises practical concerns for real-world deployment.

2. The method relies on a curated interleaved reasoning corpus for supervised training. It is not fully clear how generalizable these textual–visual “thought” sequences are. For example, if the SFT dataset is biased or lacks certain complex reasoning patterns, the RL stage will be unable to discover them.

**Questions:**

1. The qualitative examples focus on successes. Could you provide and analyze a few failure cases? Specifically, are there types of problems where the AD-Loop consistently fails or even degrades performance compared to simpler reasoning strategies?
2. Could you provide more detail on the heuristics used to "reorganize" and "synthesize" the AD-Loop traces from existing datasets (Sec 3.2)? How did you ensure the quality and logical coherence of these generated traces?
3. What is the typical computational overhead (e.g., latency, FLOPs) of invoking the AD-Loop for a complex query compared to a standard generation? How does the number of interleaved steps affect this cost?

---

> ### Author Response · Authors · 2025-11-22
> **Response Part-1/2**
>
> We sincerely thank Reviewer #h8QD for the detailed comments.
> Your concerns helped us significantly strengthen the clarity and practicality of the proposed framework.
> Below, we provide thorough, point-by-point responses.
>
> ----
>
> **Q1. The framework introduces significant complexity in both training and inference. It requires a specialized two-phase training (including an RL stage), and at runtime the model must perform multiple analyze–draft iterations per query. This likely incurs substantial computational cost and latency compared to standard one-pass models. The paper does not discuss inference speed or resource requirements, which raises practical concerns for real-world deployment.**
> **A**: We appreciate this important concern and would like to clarify that AD-Loop does not introduce substantial inference overhead for the following reasons.
>
> *(1) Adaptive invocation rather than mandatory usage*. The model learns when visual thoughts are unnecessary and defaults to a standard one-pass inference path. As shown in Figure 6, a portion of inputs where AD-Loop is never triggered, resulting in identical inference cost to the baseline model.
>
> *(2) Visual drafts are compact*. Visual thoughts are represented as at most 16 latent tokens, rather than full pixel-level images. For comparison, generating a 384*384 image in BAGEL or Janus-Pro requires 576 image tokens, which is more than 36x larger than our latent representation. This keeps per-iteration overhead lightweight and does not significantly increase computational cost.
>
> *(3) Stage-2 RL affects only training and imposes no inference-time computation*. Stage-2 reinforcement learning is performed exclusively during training and introduces no additional computation at inference time. The runtime behavior remains identical to standard decoding except when the model chooses to perform an AD-Loop iteration.
>
> *(4) Measured latency remains low in practice*. Across benchmarks, we observe an average inference latency of 1.12× the baseline BAGEL with thinking mode. Even with 3–4 AD-Loop iterations, the additional token budget corresponds to only 48–64 extra latent tokens, which remains minimal in practice.
>
> Overall, AD-Loop is not computationally heavy because it is adaptive, uses compact latent visual thoughts, applies RL only during training, and exhibits empirically low inference overhead.
>
>
>
> ---
>
>
>
>
> **Q2. The method relies on a curated interleaved reasoning corpus for supervised training. It is not fully clear how generalizable these textual–visual “thought” sequences are. For example, if the SFT dataset is biased or lacks certain complex reasoning patterns, the RL stage will be unable to discover them.**
> **A**: This is an excellent point. We first clarify that when constructing the SFT corpus, we explicitly aimed to mitigate bias by covering diverse domains (e.g., math, games, general QA) and both understanding and generation paradigms. The textual–visual thinking trajectories were derived based on existing datasets that provide detailed reasoning annotations, ensuring correctness and consistency rather than arbitrary synthesis.
>
> Importantly, the SFT stage serves only as an initial scaffold, not as the full reasoning space. Stage-1 teaches the model the format of interleaved thoughts and the general alternation structure, but does not constrain the content or strategies used during reasoning.
> Stage-2 RL then extends beyond the SFT distribution. Because the reward is assigned based solely on the final answer or generated image, the policy is not supervised to reproduce specific thought sequences. Instead, RL operates in a closed-loop manner and can discover new reasoning patterns whenever they lead to higher rewards. In practice, this allows the model to go beyond any biases or omissions in the SFT corpus.

---

> ### Author Response · Authors · 2025-11-22
> **Response Part-2/2**
>
> **Q3. The qualitative examples focus on successes. Could you provide and analyze a few failure cases? Specifically, are there types of problems where the AD-Loop consistently fails or even degrades performance compared to simpler reasoning strategies?**
> **A**:
> Thank you for raising this important question. We agree that analyzing failure cases is essential for understanding the limitations of AD-Loop. We have included representative examples in Appendix Figures 16 and 17, and summarize the main observations below.
>
> On the understanding side, the model may occasionally over-draft on simple queries. For very basic attribute questions, the introduction of additional reasoning steps can lead to unnecessary elaboration and, in some cases, incorrect predictions.
> Similarly, although the model shows improvements on visually grounded mathematical tasks, such as MathVista, it still struggles on problems that require strict symbolic reasoning, where applying AD-Loop may introduce deviations from the optimal logical path.
>
> On the generation side, the model continues to face difficulties in rendering fine-grained visual details, including scene text, subtle body parts (e.g., fingers), and numerically precise elements. A plausible explanation is that the compact latent visual representations may under-encode such fine details. We anticipate that incorporating more detailed feedback into the Stage-2 RL reward, encouraging the model to attend to fine-grained latent visual thoughts, could further mitigate these issues in future work.
>
>
>
>
>
> ---
>
> **Q4. Could you provide more detail on the heuristics used to "reorganize" and "synthesize" the AD-Loop traces from existing datasets (Sec 3.2)? How did you ensure the quality and logical coherence of these generated traces?**
> **A**: We appreciate this question. Additional details on the data construction process are provided in Appendix F, along with corresponding visualizations. Here, we clarify how the quality and logical coherence of the synthesized AD-Loop traces are ensured.
>
> First, many of the reorganized traces are derived from existing multimodal CoT datasets that already contain high-quality reasoning structures. For example, Visual-CoT includes intermediate reasoning annotations with bounding boxes that explicitly indicate the critical visual regions required for solving the task. We convert these annotated regions into explicit visual thoughts by zooming into and cropping the corresponding areas, ensuring that the derived visual steps remain semantically aligned with the original rationale.
>
> Second, for datasets constructed by us, we generate the traces in two steps. We first produce self-reflective textual critiques and decompose them into subgoals (e.g., identifying key objects or spatial dependencies). We then synthesize corresponding visual hypotheses using a high-quality model and retain only logically consistent visual thoughts. The resulting AD-Loop traces are shown in Appendix Figure 9.
>
> Together, these procedures ensure that the synthesized traces are both semantically grounded and logically coherent.
>
>
>
> ---
> **Q5. What is the typical computational overhead (e.g., latency, FLOPs) of invoking the AD-Loop for a complex query compared to a standard generation? How does the number of interleaved steps affect this cost?**
> **A**:  Thank you for the question. The computational overhead of invoking the AD-Loop is modest and remains tightly controlled in practice. A single visual-draft iteration incurs approximately +7% FLOPs due to the addition of latent visual tokens. In most cases, the model selects in 3-4 AD-Loop iterations. As a result, the average end-to-end inference latency is around 1.12× that of the backbone UVLM (BAGEL) across benchmarks.  The additional token budget corresponds to only 48–64 extra latent tokens.
> Importantly, the memory footprint remains unchanged because latent visual thoughts are represented as compact token embeddings rather than pixel-level renderings, and no diffusion or external vision module is invoked during thinking.

---

> ### Author Response · Authors · 2025-11-27
>
> Dear reviewer h8QD,
>
> Thank you again for your thoughtful review.
> If there are any remaining concerns you’d like us to clarify, we would be glad to provide further explanations.
>
> And if our response has successfully addressed your previous concerns, we kindly hope you might consider updating that in your final assessment. We really appreciate your time and feedback!

---

### Official Review · Reviewer_QySp · 2025-11-01

**Soundness:** 3
**Presentation:** 3
**Contribution:** 3
**Rating:** 6
**Confidence:** 3

**Summary:**

Existing approaches primarily focus on architectural unification while overlooking the importance of explicit interaction between the two capabilities during task solving.
This paper introduces the interleaved Analyzing–Drafting problem-solving loop (AD-Loop), a new think paradigm that dynamically alternates between analytic and drafting operations.

**Strengths:**

- The paper introduces a two-stage training strategy: (1) supervised learning on interleaved thought data to initialize the alternation, followed by (2) reinforcement learning to promote adaptive and autonomous control.
- The paper conducts extensive experiments and ablation studies to validate the effectiveness of the proposed method.
- AD-Loop consistently improves performance across standard benchmarks for both understanding and generation, and further showed the adaptability of the proposed method on other unified VLM.

**Weaknesses:**

- The definition of the inter-group reward in Equation (6) is unclear. What does $m$ represent? Does it indicate whether the trajectory is AD-Loop-enabled or not? Additionally, in Equation (7), the intra- and inter-reward terms on the right-hand side seem to be missing the superscript $m$?
- In the ablation study (Table 3), it is unclear whether the paper trained three additional variants corresponding to different thinking strategies and evaluated them separately, or whether they used the final model but selectively disabled certain thinking capabilities.
- In Figure 4, what does unified-R1 refer to? Is it AD-Loop?
- Although Figure 5 presents examples of latent visual thoughts, it would be beneficial if the paper also included qualitative examples illustrating the complete thinking trajectories along with visualizations of the generated latent visual thoughts.

**Questions:**

- When constructing the dataset, did you control the number of visual drafts generated during the thinking process? What were your design choices and constraints here?
- In the final training loss (Equation 2), it appears that all loss components are simply summed together without weighting. Could you clarify whether any weighting scheme was considered or tested?

---

> ### Author Response · Authors · 2025-11-22
> **Response Part-1/2**
>
> We sincerely thank Reviewer #QySp for the positive assessment and the constructive questions.
> Below we provide detailed responses to each raised concern.
>
> -----
>
> **Q1. The definition of the inter-group reward in Equation (6) is unclear. What does it represent? Does it indicate whether the trajectory is AD-Loop-enabled or not? Additionally, in Equation (7), the intra- and inter-reward terms on the right-hand side seem to be missing the superscript ?**
>
> **A**: Thank you for pointing out this important clarity issue.
> Regarding the inter-group reward in Equation (6), for each input, we compare the best trajectory sampled from the AD-Loop group (“+”) with the best trajectory from the non–AD-Loop group (“–”).
> The group achieving the higher maximum reward receives the inter-group reward.
> This design enables the policy to adaptively learn when AD-Loop is beneficial, which is essential for allowing the model to autonomously decide whether to invoke AD-Loop at inference time.
>
> You are correct that there are some issues in Equation (7). We have corrected the notation accordingly in the revised version of the paper.
>
> ----
>
> **Q2. In the ablation study (Table 3), it is unclear whether the paper trained three additional variants corresponding to different thinking strategies and evaluated them separately, or whether they used the final model but selectively disabled certain thinking capabilities.**
> **A**: Thank you for asking. To ensure a fair comparison, we trained three separate model variants. All variants were initialized from the same pretrained checkpoint, trained on the same data, and evaluated independently under identical settings. Specifically:
>
> -  Analyzing-only thinking ($T$): Train the model to emit only textual thoughts.
>
> - Textual–Explicit visual Interleaving ($T+I$): Train the model with only textual and explicit pixel-level visual interleaved thoughts.
>
> - Textual–Implicit visual Interleaving ($T+\widetilde{I}$): Train the model with only textual and implicit visual interleaved thoughts.
>
>
>
>
> ----
>
>
> **Q3. In Figure 4, what does unified-R1 refer to? Is it AD-Loop?**
> **A**: Thank you for pointing this out. Yes, unified-R1 refers to the AD-Loop mechanism. We refine the terminology in the revised version.
>
>
>
> ----
>
> **Q4. Although Figure 5 presents examples of latent visual thoughts, it would be beneficial if the paper also included qualitative examples illustrating the complete thinking trajectories, along with visualizations of the generated latent visual thoughts.**
> **A**: Thank you for the helpful suggestion. We would like to clarify that a complete thinking trajectory is already provided in *Figure 3*, where the `[V-T]` markers indicate the occurrence and placement of latent visual thoughts. As noted, latent visual thoughts are represented as latent tokens rather than explicit images. In *Figure 5*, the visualizations are shown only to illustrate how these latent tokens correspond to clustered regions in the original image (left side of each subfigure), demonstrating that the representations are semantically coherent rather than random or average pooling. The detailed procedure is described in Section 3.2. Additionally, we have included full trajectories for different task types in Appendix Figures 10–12.

---

> ### Author Response · Authors · 2025-11-22
> **Response Part-2/2**
>
> **Q5. When constructing the dataset, did you control the number of visual drafts generated during the thinking process? What were your design choices and constraints here?**
> **A**: Thank you for the question. Yes, the number of latent visual thoughts was explicitly controlled during dataset construction, and the choice varies across datasets depending on the annotation format and task structure. Below, we clarify the design decisions for each dataset:
>
> - *CMoT*: This dataset contains four task types with different supervision forms. The average number of latent visual thoughts is approximately 3.
>
> - *Visual-CoT*: Since intermediate reasoning annotations with bounding boxes are explicitly provided, each annotated region is treated as a latent visual thought, resulting in an average of 2 per trajectory.
>
> - *CoF*: The dataset includes annotated focus regions. We derive explicit visual thoughts by cropping these regions, yielding an average of 3 per sample.
>
> - *Visual Spatial Planning (VSP)*: Visual thoughts correspond to the visualization of each movement step, with an average of 3 latent visual thoughts.
>
> - *GoT-T2I*: We treat the annotated bounding boxes of salient objects as latent visual thoughts, resulting in an average of 4.
>
> - *Inter-T2I*: For our constructed dataset, we generate visual hypotheses based on sub-goals extracted from the prompts and retain only the first occurrence of each object to avoid duplication. The average number of latent visual thoughts is 4.
>
> We have summarized these statistics in the revised Appendix Section F and added them to Table 5 for clarity.
>
>
>
>
> ----
> **Q6. In the final training loss (Equation 2), it appears that all loss components are simply summed together without weighting. Could you clarify whether any weighting scheme was considered or tested?**
> **A**: Thank you for the careful observation. After re-examining our implementation, we confirm that a weighting scheme was indeed used in Equation (2). In the revised version, we explicitly introduce the coefficient $\alpha$. We conducted a hyperparameter search over
> $\alpha \in \{0.1,0.2,0.4,0.5,0.8,1.0\}$ and observed minimal variation in performance across this range. Finally, we therefore set $\alpha=1$ in all experiments, which aligns with our design intuition: latent visual thoughts and textual thoughts serve complementary roles and contribute equally to the optimization objective.
> In addition, we corrected the notation in Equation (7) to include the weighting coefficient $\gamma$, which balances the intra-group and inter-group advantages. We set $\gamma=1$ in our experiments after a preliminary study. All updated settings are provided in Appendix G.1.

---

> > ### Comment · Reviewer_QySp · 2025-11-27
> >
> > Thank you for the detailed response. It has addressed most of my concerns. I would like to adjust my score to a 7 based on the clarifications. However, since the system does not offer a score of 7, I will keep my current score. I would nevertheless like to note that my intended score is 7 and recommend the paper for acceptance.

---

> > > ### Author Response · Authors · 2025-11-27
> > >
> > > Thank you for your positive feedback and for letting us know your intended score. We truly appreciate your recognition of our work and are glad that our clarifications have addressed your concerns.
> > >
> > > Since the system does not provide a score of 7, we kindly wonder whether it might be possible to reflect your intended rating by adjusting the score to an 8. Of course, we fully respect your judgment, and if there are any remaining questions or concerns, we would be more than happy to discuss them further.

---

### Author Response · Authors · 2025-11-22
**To All Reviewers**

We sincerely thank all reviewers for the thoughtful evaluation and highly constructive feedback. We are encouraged by the strong positive reception (scores of 8, 8, and 6), and we especially appreciate the detailed critique from Reviewer #h8QD (score 4), which helped us clarify important aspects and strengthen the manuscript. Below, we provide an expanded and rigorous set of responses that address every concern in depth.

Our submission introduces the Interleaved Analyzing–Drafting problem-solving Loop (AD-Loop), a novel thinking paradigm that enables the model to dynamically alternate between understanding and generation, thereby fostering an organic synergy that enhances overall problem-solving capability.

We highlight the core motivations and contributions clarified across the rebuttal:

- **Novelty of AD-Loop**:
AD-Loop is not merely another form of visual-augmented CoT. Instead, it introduces an iterative bidirectional thinking loop where textual reasoning controls and interprets latent visual drafts, while visual drafts reshape subsequent textual analysis. This also enables a flexible, interleaved use of the understanding and generation abilities of the UVLM in the problem-solving loop.

- **Latent Visual Thoughts**:
AD-Loop relies on a new representation: low-dimensional latent visual thought tokens, learned through clustering of generator-side image representations. These tokens provide a visual abstraction far more coherent and computationally efficient than pixel-level representations.

- **Adaptive Reasoning via Group-Relative RL**:
The proposed RL design enables the model to choose whether AD-Loop should be invoked for a given input. This adaptiveness is critical for efficiency and generalization, and is completely absent from prior approaches that employ fixed or heuristic alternation.

- **Practical Efficiency**:
Despite its conceptual sophistication, AD-Loop has surprisingly low inference overhead due to latent thinking representations and adaptive invocation. The full model averages around 1.12× backbone baseline (i.e., BAGEL w/ think) inference cost across all benchmarks.

- **Robust Empirical Benefits**:
AD-Loop improves both understanding and generation tasks across 10+ benchmarks.

We have revised the manuscript to include clearer explanations, additional discussions, and representative failure cases in response to the reviewers’ comments.

If there are any remaining questions, we would be more than willing to provide further clarification.

---

### Author Response · Authors · 2025-12-01

Dear ACs and PCs,

We sincerely appreciate the reviewers' recognition of our work (reflected in the positive scores: 8, 8, 6, and 4) and their dedicated efforts in providing constructive and thoughtful feedback.
Below, we present a factual summary of the reviewers' initial concerns as well as the outcomes after the rebuttal and discussion phases. We respectfully hope that these points will be taken into consideration in your final decision.


## Initial Concerns and Rebuttal

The reviewers' initial concerns centered on the following aspects:

**1. Need for additional differentiating justification for AD-Loop compared with existing methods.**
Reviewers requested further elaboration on the fundamental differences between our proposed AD-Loop and prior methods. In response, we provided a detailed explanation from three complementary perspectives, *objective, modeling, and learning*, and incorporated the corresponding clarifications into the revised manuscript.

**2. Need for a more thorough error case study.**
The reviewers suggested including failure case analysis and hoped to see interesting findings that could inspire future research. During the rebuttal phase, we conducted additional experiments, added several representative failure cases to the revised manuscript, and highlighted our observations.

**3. Concerns about efficiency.**
The reviewers raised questions regarding the complexity and efficiency of the proposed framework. To address this, we included additional experiments during the rebuttal phase to validate the efficiency of our method and provided more detailed explanations in the responses.

**4. Need for improved presentation clarity.**
Some reviewers commented that parts of the presentation, such as certain formulas and descriptions of the proposed mechanism, were unclear. We refined the explanations, improved the clarity of the corresponding sections, and highlighted all revisions in the updated manuscript.



## Discussion Progress

After the initial rebuttal, several reviewers provided positive feedback. We summarize the discussion progress below:

- Reviewer `#QySp` indicated that their concerns had been resolved. They explicitly stated their intention to raise the score and recommended the paper for acceptance.

- Reviewer `#h8QD`, despite the initial score of 4, acknowledged that our work represents a new paradigm, well-motivated method and consistent improvements. We responded to all of their concerns one-by-one. However, as of November 27, the reviewer has not engaged in the discussion phase, and we sincerely hope that our responses have adequately addressed their concerns.

- Reviewers `#PeBS` and `#tVfz` both gave strong initial positive scores (8). Unfortunately, there was no further discussion from them before November 27.


## Conclusion

Overall, we have carefully and thoroughly addressed every concern and question raised by the reviewers. All new experiments, clarifications, and expanded discussions have been fully incorporated into the revised manuscript.

We respectfully ask that these efforts and clarifications be taken into consideration during your final assessment.

Sincerely,
Authors

---

### Meta-Review · Area_Chair_td7j · 2026-01-03

**Summary:**

The Unified Visual Language Model (UVLM) addresses understanding and generation within a single framework. However, existing approaches often inadequately design their interaction, treating them as parallel skills. Therefore, this paper proposes the "interleaved analyzing–drafting problem-solving loop (AD-Loop)," which alternates dynamically between analysis and drafting. By alternating between textual and visual thinking, the AD-Loop iteratively refines comprehension and output, promoting synergistic effects. Learning occurs in two stages: initialization via supervised learning using alternating thinking data and reinforcement learning to enhance autonomous control. Experiments confirm that the AD-Loop consistently improves comprehension and generation on standard benchmarks and generalizes to diverse UVLM architectures. Furthermore, visualizations demonstrate the effectiveness of implicit visual reasoning.

Reviewers raised the following points: (1) novelty and distinction from related work, (2) data dependency and generalizability, (3) insufficient qualitative analysis, (4) computational cost and practical concerns, and (5) ambiguity in definitions, notation, and explanations. However, the authors' responses reinforced their explanations, particularly regarding novelty and differentiation from related work. The AC judges that many concerns were addressed. The AC recommends Accepting this paper because it presents a new thinking paradigm that is expected to contribute to the community.

**Reviewer Concerns:**

The reviewers primarily raised the following concerns:
- Novelty and distinction from related work (insufficient definition of the "synergistic effect" in AD-Loop)
- Concerns regarding data dependency and generalizability
- Insufficient qualitative analysis (presentation of thought processes and failure cases)
- Computational cost and practical implementation concerns
- Ambiguity in definitions, notation, and explanations

The AC considers these concerns resolved by the authors' responses.

**Reviewer Scores:**

Reviewer QySp is satisfied with the authors' responses and states that while the score is actually 6, a score of 7 would also be acceptable.

Reviewer h8QD gave a score of 4 and did not participate in the final discussion. This reviewer's concerns were that the proposed method introduces significant complexity in both learning and inference, and concerns about data dependency and generalizability. The authors' responses were appropriate, and AC believes this reviewer may ultimately raise their score.

Reviewer PeBS gave a score of 8 and did not participate in the final discussion. This reviewer's primary concern is the novelty of the proposed method. Since the authors have responded appropriately to this concern, the AC believes the score is unlikely to drop significantly from 8.

Reviewer tVfz scored 8 and did not participate in the final discussion. This reviewer's primary concern was the difference from related work. Since the authors have responded appropriately to this concern, the AC believes the score is unlikely to drop significantly from 8.

---

### Decision · Program_Chairs · 2026-01-26

Accept (Poster)